# Compressing Tree Ensembles through Level-wise Optimization and Pruning

**Laurens Devos** [* 1 2]  **Timo Martens** [* 1 2]  **Deniz Can Oruç** [1 2]  **Wannes Meert** [1 2]  **Hendrik Blockeel** [1 2]  **Jesse Davis** [1 2]

## Abstract

Tree ensembles (e.g., gradient boosting decision trees) are often used in practice because they offer excellent predictive performance while still being easy and efficient to learn. In some contexts, it is important to additionally optimize their size: this is specifically the case when models need to have verifiable properties (verification of fairness, robustness, etc. is often exponential in the ensemble's size), or when models run on battery-powered devices (smaller ensembles consume less energy, increasing battery autonomy). For this reason, compression of tree ensembles is worth studying. This paper presents LOP, a method for compressing a given tree ensemble by pruning or entirely removing trees in it, while updating leaf predictions in such a way that predictive accuracy is mostly unaffected. Empirically, LOP achieves compression factors that are often 10 to 100 times better than that of competing methods.

## 1. Introduction

Ensembles of decision trees, a.k.a. decision forests, are frequently used in practice because of their ease of training, excellent overall performance, and high efficiency. Their training requires little or no hyperparameter tuning, the computational cost of both training and prediction is very small, and for tabular data, they are usually among the best performing models (Grinsztajn et al., 2022).

This paper is about compressing decision forests: reducing their size as measured by, e.g., the total number of leaves in the forest, while maintaining their accuracy. In the light of the above successes, one might wonder whether this is worthwhile: if forests are already so efficient, can we gain

---
[*]Equal contribution  [1]KU Leuven Department of Computer Science, Leuven, Belgium [2]Leuven.AI, KU Leuven Institute for Artificial Intelligence, Leuven, Belgium. Correspondence to: Jesse Davis <jesse.davis@kuleuven.be>, Hendrik Blockeel <hendrik.blockeel@kuleuven.be>.

*Proceedings of the 42nd International Conference on Machine Learning*, Vancouver, Canada. PMLR 267, 2025. Copyright 2025 by the author(s).

much from reducing their size? The answer is yes, for multiple reasons: (1) Due to their efficiency, forests are very suitable for use on embedded or battery-powered devices (Fan et al., 2013; Donos et al., 2015; Daghero et al., 2021; Lauwereins et al., 2015). A smaller forest has a smaller memory footprint and consumes less energy per prediction (Buschjäger & Morik, 2023). On battery-powered devices, this can have a substantial effect on battery autonomy. (2) Forest size is relevant when models must be verifiable, for instance, in the context of fairness or robustness against adversarial attacks (Devos et al., 2021b;a). Verification methods for decision forests tend to scale exponentially in the size of the forest (Kantchelian et al., 2016; Devos et al., 2023). Therefore, any size reduction has an important effect on verifiability.

While one could try to construct smaller forests from the start, the success of existing methods such as Random Forests or XGBoost is an argument for not changing the method of growing the forest, but reducing its size afterwards (Ren et al., 2015; Buschjäger & Morik, 2023). In this formulation, the original forest provides a reference point in terms of predictive performance. This enables designing compression approaches that more explicitly trade off predictive accuracy versus other aspects of performance, such as memory footprint, energy consumption, and verifiability (all of which correlate with size).

Multiple methods for reducing the size of forests have been proposed. There has been extensive research on removing component models from ensembles (Tsoumakas et al., 2009). More recently, methods specific to forests have been proposed. These usually optimize the values stored in the leaves, with size reduction as a possible side-effect rather than the primary goal (e.g, Ren et al., 2015), though several approaches explicitly add pruning as an objective (Buschjäger & Morik, 2023; Liu & Mazumder, 2023a). Generally, all these methods try to find an optimum for some cost function that trades off size versus accuracy.

In this paper, we propose a novel method called LOP for compressing decision forests. LOP tries to find the smallest forest whose predictive accuracy is still within a user-provided margin to that of the original forest. Like other methods, it starts from a given forest. In contrast to existing methods that restrict what can be pruned, LOP can prune any

(sub)tree in the forest. It proceeds level-wise, from the root towards the leaves. At each level, it formulates an optimization problem that tries to prune subtrees rooted at that level while updating the values in the remaining leaves to avoid a drop in accuracy. Empirically, LOP can find much smaller forests than existing methods: sometimes by a factor 100 or more, if a small loss in accuracy ($< 0.5\%$) is acceptable; and often by a factor 2-3 when comparing forests with the same accuracy. This substantially increases the applicability of forests in contexts where verifiability, robustness, battery autonomy, etc. matter.

## 2. Preliminaries

**Decision trees.** Decision trees exist in many variants (Blockeel et al., 2023). We take the following view: A decision tree is a tree-shaped representation of a function $\mathcal{X} \to \mathcal{Y}$; internal nodes of the tree are labeled with a test that can be performed on instances; there is an outgoing edge for each possible outcome of the test; leaves are labeled with a value $\hat{y} \in \mathcal{Y}$. A tree associates each instance $x$ with a value $\hat{y}$ by sorting $x$ down the tree according to the outcome of the tests until a leaf is reached, and returning the value stored in that leaf. Here, we assume $\mathcal{Y} = \mathbb{R}$: this covers regression trees but also binary classification trees by using a threshold on $\hat{y}$ to decide the predicted class.

**Decision forests.** Decision forests are ensembles of decision trees. Learning algorithms for them include popular algorithms such as Random Forests (Breiman, 2001) and gradient boosted decision trees (Friedman, 2001), of which XGBoost (Chen & Guestrin, 2016) is a well-known example. Forests are typically constructed iteratively, growing one tree at a time using a standard tree learner and adding it to the forest until that contains a predefined number of trees. A forest $\boldsymbol{T} = \{T_1, T_2, \ldots T_M\}$ makes predictions by combining the predictions of the individual trees $T_m$. The combination rule can be standard voting for classification or averaging for regression, but more generally it is a linear combination of the individual predictions: $\boldsymbol{T}(x) = \sum_m w_m T_m(x)$ where both the $T_m$ and the $w_m$ are learned.

A tree can be written as an additive model: $T(x) = \sum_j \mathbb{1}_j(x)\hat{y}_j$ with $\hat{y}_j$ the value stored in the $j$'th leaf, and $\mathbb{1}_j$ is 1 if $x$ gets sorted into the $j$'th leaf and 0 otherwise. Hence, the entire forest can be written in an additive form:

$$\boldsymbol{T}(x) = \sum_m w_m \sum_j \mathbb{1}_{m,j}(x)\hat{y}_{m,j} = \sum_{m,j} w_m \mathbb{1}_{m,j}(x)\hat{y}_{m,j}$$

where $m$ indexes trees and $j$ leaves in a tree. The notation can be simplified to

$$\boldsymbol{T}(x) = \sum_{k=1}^{L} \mathbb{1}_k(x)v_k$$

where $k$ indexes all $L$ leaves in the forest and each $v_k$ equals some $w_m \hat{y}_{m,j}$; and ultimately to

$$\boldsymbol{T}(x) = s(x) \cdot v,$$

the dot product of an $L$-dimensional vector $v$ with a binary vector $s(x)$ that indicates which leaves $x$ is sorted into (i.e., the $k$'th component of $s(x)$ is $\mathbb{1}_k(x)$).

**Postprocessing decision forests.** Forests are learned in a greedy manner: tree $T_m$ is learned in the context of the partial forest $\{T_1, T_2, \ldots, T_{m-1}\}$, but the tree that seems most useful at that point may turn out less useful in the context of the forest as a whole. Furthermore, because the number of trees is predefined, more trees may be added than strictly needed to achieve a certain level of accuracy.

This has prompted researchers to investigate whether forests can be further optimized after learning them. Forests can often be simplified without loss of predictive accuracy by removing trees that eventually turn out to be redundant (Margineantu & Dietterich, 1997; Buciluă et al., 2006; Tsoumakas et al., 2009; Lu et al., 2010; Zhang et al., 2006). These approaches aim primarily at reducing the forest size, ideally at no loss in accuracy.

A different type of postprocessing approach (e.g., Ren et al., 2015) is based on changing the values predicted in the leaves of the trees, which is often referred to as *leaf refinement*. By default, the values inserted into $T_m$'s leaves are optimal in the context in which $T_m$ is learned. They may not be optimal in the context of the whole forest. In fact, it is easy to show that in general, individual trees may have to make suboptimal predictions for the ensemble to make optimal ones (see Appendix A).

## 3. The LOP Algorithm

We introduce Level-wise Optimization and Pruning (LOP), a novel algorithm which tackles the following problem:

**Given:** A previously learned tree ensemble $\boldsymbol{T}$ and a dataset $\{(x_i, y_i)\}_{i=1}^{N}$ of $N$ examples.

**Do:** Compress $\boldsymbol{T}$ into a (much) smaller ensemble $\boldsymbol{T}'$ such that there is a minimal difference in predictive performance between $\boldsymbol{T}$ and $\boldsymbol{T}'$.

The core idea underlying LOP is that, starting with the root nodes (level 0), it processes the ensemble in a level-by-level manner. At each level, it formulates an optimization problem that simultaneously attempts to prune (sub)trees and optimize the leaf values. This problem is global in that it involves *all nodes* in the ensemble at the considered level.

To illustrate the intuition behind the approach, consider the simple ensemble shown in Figure 1a and suppose the

algorithm is processing level 1. If the leaf values $v_{11}$, $v_{12}$ and $v_{13}$ under the node HEIGHT $< 190$ in $T_2$ were all equal to $v$, then clearly that subtree could be replaced by a leaf with value $v$. To encourage such compression, LOP jointly optimizes the leaf values under each node $n$ at this level by learning a transformation $c_n v_k + b_n$ where $c_n$ and $b_n$ are learnable parameters. LOP then runs an optimization problem that finds $c_n$, $b_n$ such that the new forest (with leaf values of the form $c_n v_k + b_n$) best fits the training data. During this optimization, LOP applies the sparsity inducing L1 regularization on on $c_n$, yielding two cases:

1. $c_n = 0$: the subtree rooted at $n$ is pruned and re- placed by a leaf node with value $b_n$. Figure 1b shows this for our running example where the internal node HEIGHT $< 190$ is replaced by $b_4$,

2. $c_n \neq 0$: the values of all leaf nodes under $n$ are updated to be $c_n v_k + b_n$. This is shown for the nodes under node AGE $< 50$ in Figure 1b.

For leaf nodes (i.e., $v_1$ and $v_8$ in this example), the $c_n$ value is already zero and the above simply entails learning a new, globally optimal leaf value.

Next, we formalize more rigorously how LOP works.

### 3.1. Formulating the optimization problem

When formulating the optimization problem for level $l$, there is one complicating factor that must be considered: some trees may have leaf nodes at a higher level (i.e., some level $< l$). For example, in Figure 1 this is the case for both trees at level 2. Because these leaves affect the prediction for each example, they must also be considered in the optimization problem. Moreover, the values of such leaves will also need to be updated to reflect any changes to the ensembles at level $l$. Hence, for level $l$ we define the the set of **active nodes** as the union of all nodes (both internal and leaf) at level $l$ together with all leaf nodes with a level $< l$. Below, we use $n$ to index active nodes, and $k$ to index leaf nodes.

Before the transformation, the prediction for an instance $x$ equals $s(x) \cdot v$. Changing all $v_k$ into $c_{n(k)} v_k + b_{n(k)}$, with $n(k)$ the active node above the $k$'th leaf, turns $v$ into $(c' \odot v + b')$, where $\odot$ is componentwise multiplication and the vectors $c'$ and $b'$ associate the right $c_n$ and $b_n$ value with each $v_k$ (that is, $b'_k = b_{n(k)}$ and $c'_k = c_{n(k)}$). Thus, the cost function minimized at each level is

$$\sum_{i=1}^{N} \ell(s(x_i) \cdot (c' \odot v + b'), y_i) + \alpha \sum_n |c_n| \quad (1)$$

with $\ell$ a loss function.

Practically, this optimization problem can be written as

$$\min_\theta \ell(X\theta, y) + \alpha r(\theta) \quad (2)$$

where $X$ is the data matrix, $\theta$ contains the parameters to be optimized, and $r$ is a regularization term (see 3.2). Let us denote with $a$ the number of active nodes at the current level. For now, assume the set of active nodes contains no roots or leaves. $\theta$ is then a $2a$-dimensional vector

$$\theta = \begin{bmatrix} c_1 & b_1 & c_2 & b_2 & \dots & c_a & b_a \end{bmatrix}^T.$$

$X$ represents the dataset in terms of the nodes and leaves that each instance will be sorted into, and the corresponding leaf values. $X$ can be computed as $SB$ with $S$ an $N \times L$ matrix with as $i$'th row $s(x_i)$, and $B$ an $L \times 2a$ matrix that has on row $k$ the values $v_k$ and 1 in columns $2n(k) - 1$ and $2n(k)$. Basically, $S$ maps instances to leaves and $B$ links the corresponding leaf values to the right $c$, $b$ parameters. It is easily verified that $X\theta$ is then a column matrix whose $i$'th row equals $s(x_i)(c' \odot v + b')$. The general case where active nodes can be leaves is obtained by simply omitting from $\theta$ the $c_n$ component for leaves (leaves need not be pruned any further) and dropping the corresponding columns in $B$. Level 0 is another special case: pruning at level 0 removes the tree entirely, so no bias term is used at that level.

*Example* 3.1. The ensemble shown in Figure 1, top, has 8 leaves. Counting leaves from left to right, assume that the first three instances in the training set are sorted into leaves 2 and 7; 1 and 6; and 4 and 5. $SB\theta$ is then:[1]

$$\begin{bmatrix} 0 & 1 & 0 & 0 & 0 & 0 & 1 & 0 \\ 1 & 0 & 0 & 0 & 0 & 1 & 0 & 0 \\ 0 & 0 & 0 & 1 & 1 & 0 & 0 & 0 \\ \vdots & \vdots & \vdots & \vdots & \vdots & \vdots & \vdots & \vdots \end{bmatrix} \begin{bmatrix} v_1 & 1 & 0 & 0 & 0 & 0 & 0 & 0 \\ 0 & 0 & v_3 & 1 & 0 & 0 & 0 & 0 \\ 0 & 0 & v_5 & 1 & 0 & 0 & 0 & 0 \\ 0 & 0 & v_6 & 1 & 0 & 0 & 0 & 0 \\ 0 & 0 & 0 & 0 & v_8 & 1 & 0 & 0 \\ 0 & 0 & 0 & 0 & 0 & 0 & v_{12} & 1 \\ 0 & 0 & 0 & 0 & 0 & 0 & v_{13} & 1 \\ 0 & 0 & 0 & 0 & 0 & 0 & v_{14} & 1 \end{bmatrix} \begin{bmatrix} c_1 \\ b_1 \\ c_2 \\ b_2 \\ c_3 \\ b_3 \\ c_4 \\ b_4 \end{bmatrix}$$

and we have

$$X = SB = \begin{bmatrix} 0 & 0 & v_3 & 1 & 0 & 0 & v_{13} & 1 \\ v_1 & 1 & 0 & 0 & 0 & 1 & v_{12} & 1 \\ 0 & 0 & v_6 & 1 & v_8 & 1 & 0 & 0 \\ \vdots & \vdots & \vdots & \vdots & \vdots & \vdots & \vdots & \vdots \end{bmatrix}$$

and

$$X\theta = \begin{bmatrix} v_3 c_2 + b_2 + v_{13} c_4 + b_4 \\ v_1 c_1 + b_1 + v_{12} c_4 + b_4 \\ v_6 c_2 + b_2 + v_8 c_3 + b_3 \end{bmatrix}$$

### 3.2. Maintaining predictive performance

There are two issues to contend with when solving our optimization problem. First, compression, i.e., pruning (sub)trees, may adversely affect the predictive performance.

---

[1]In practice, the columns/rows shown in red would be omitted and are just shown to better visualize the matrix structure.

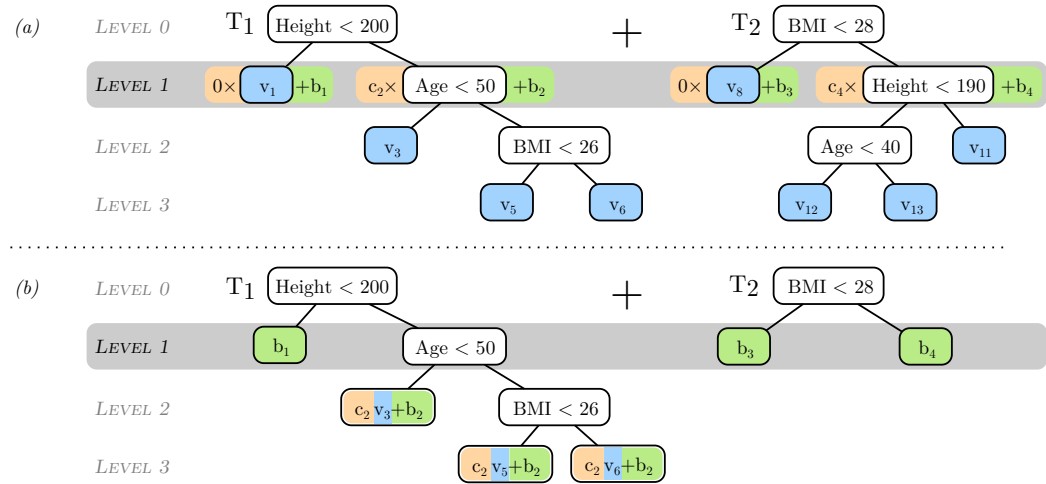

**Figure 1.** An example ensemble consisting of two trees $T_1$ and $T_2$, and the effect of applying the LOP subroutine at level 1. The leaf constants of the ensemble are denoted by $v_k$. (A) shows the original ensemble annotated with the scaling and shifting parameters $c_n$ and $b_n$ on level 1. Note that leaf nodes have no scaling factor. (B) shows the resulting ensemble after applying the LOP subroutine at level 1, assuming the case where $c_2 \neq 0$ and $c_4 = 0$. When a scaling parameter $c_n$ is nonzero, the leaf values in the subtree rooted at node $n$ are updated to $c_n v_k + b_n$, as shown for the subtree in $T_1$. For the case where the scaling parameter $c_n$ is zero, the entire subtree rooted at $n$ is removed and replaced by a leaf with value $b_n$. This is the case for the subtree in $T_2$.

For this reason, LOP imposes the constraint that the compressed model's predictive performance on a validation set must differ by less than a user-defined margin $\Delta$ from the performance of the original forest.

LOP tunes the regularization strength parameter $\alpha$ to obtain maximal compression while satisfying the constraint on the decrease in performance using a search procedure that performs consecutive halving in log space. Starting from a wide range $[10^o, 10^u]$, LOP is run with $\alpha = 10^{(o+u)/2}$ and if the resulting model satisfies the constraint, $o$ is updated to $(o+u)/2$, otherwise $u$ is. The process is repeated until the interval is narrow enough with LOP choosing the highest $\alpha$ that satisfies the constraint. Initially, $o = -3$ and $u = 4$. In practice, LOP uses balanced accuracy[2] as the performance measure for classification, and RMSE for regression, but other measures could be used.

Second, there is a risk of overfitting. The number of parameters to be fit is at most twice the total number of active nodes, which for binary trees is upper-bounded by $M2^l$ with $M$ the number of trees in the ensemble. Thus for small values of $l$, there is little risk but the chance for overfitting increases as $l$ does. LOP's validation set approach helps counter this.

### 3.3. Overall algorithmic working

Algorithm 1 shows the pseudocode for LOP. It takes two hyperparameters: the number of rounds $R$ of level-wise

---

[2]Balanced accuracy is the accuracy of the model under a uniform class distribution; it avoids disadvantages of accuracy under skewed distributions.

compression to be performed and $\Delta$ the maximum allowed loss in predictive performance compared to the original ensemble. The process of level-wise optimization is carried out for all levels starting at level $l = 0$ (root) and ending at the maximum depth of the deepest tree in the ensemble. When $l = 0$, setting $c_n = 0$ effectively removes a whole tree from the forest. As $l$ increases, the approach considers pruning ever smaller subtrees.

The level-wise procedure is repeated $R$ times. As compression progresses, subtrees that were not prunable in a first round may become so in a subsequent round.

---

**Algorithm 1** Level-by-level compression of tree ensembles

**Input:** Ensemble $\boldsymbol{T}$, training set $D = \{(x_i, y_i)\}_{i=1}^N$ validation set $D_v = \{(x_i, y_i)\}_{i=1}^{N_v}$

**Settings:** Number of rounds $R$, maximum loss $\Delta$ in validation set performance

1: $\boldsymbol{T}' \leftarrow \boldsymbol{T}$
2: **for** $r \leftarrow 1..R$ **do**
3:    **for** $l \leftarrow 1..d$, with $d$ the maximum depth of $\boldsymbol{T}'$ **do**
4:       Construct $X = SB$ (see Section 3.1)
5:       Find largest $\alpha$ for which performance loss on $D_v$ is no larger than $\Delta$.
6:       Fit $\theta$ using Equation 2 using the selected $\alpha$.
7:       Update $\boldsymbol{T}'$: for all $n$ on this level, update all $v_k$ under $n$ to $c_n v_k + b_n$, making $n$ a leaf if $c_n = 0$
8:    **end for**
9: **end for**
10: **return** pruned ensemble $\boldsymbol{T}'$.

---

## 4. Related Work

Many methods have been proposed for reducing the size of ensembles (Tsoumakas et al., 2009). They usually perform a search through the space of subsets of the ensembles, trading off accuracy versus the number of component models. These methods are generic in the sense that they can be used on many different types of ensembles, not just ensembles of decision trees. A few methods have been proposed specifically for forests, making use of the fact that component models are known to be trees. Some of these methods have been proposed in a knowledge distillation context: these methods prioritize simplicity over accuracy. For instance, Liu & Mazumder (2023b) and Hara & Hayashi (2018) respectively aim for ensembles that contain at most 15 or 10 leaves in their experiments, accepting any accuracy loss (they minimize but do not bound it). As these methods serve a different purpose than LOP, they are less relevant here. Among other, more closely related methods, the following three are of particular interest.

**ForestPrune (FP):** Liu & Mazumder (2023a) propose an optimization approach in which decision variables indicate at what level each tree in the forest should be cut. They minimize training set loss plus a regularization term that rewards solutions with smaller average tree depth. While different trees can be cut at different levels, within each tree the cut depth is constant. This is an important difference with LOP, which can prune individual subtrees at different depths within the same tree. By cutting at one level inside each tree, ForestPrune loses a crucial aspect of trees, namely, that some subtrees can be deeper than others: a tree can partition the input space in a finer-grained manner in some areas, and in a coarser manner elsewhere. A second difference is that LOP refines leaf values whereas ForestPrune does not. Due to these two differences, LOP explores ensembles that are not in ForestPrune's search space.

**Global Refinement (GR):** Ren et al. (2015) point out that the leaf values in trees are not necessarily optimal in the context of the forest. After the full ensemble has been learned, they collectively fine-tune all the leaf values simultaneously using a single optimization problem. Their method reformulates the forest in the form $\boldsymbol{T}(x) = s(x) \cdot v$ and solves the following minimization problem:

$$\min_v \frac{1}{N} \sum_{i=1}^{N} \ell(s(x_i) \cdot v, y_i) + \alpha r(v), \qquad (3)$$

Collecting the $y_i$ in one vector $y$ and the $s(x_i)$ in a matrix $S$ whose $i$'th row equals $s(x_i)$, and defining the loss over a dataset as the sum of its instances' losses, this can also be written as

$$\min_v \ell(Sv, y) + \alpha r(v).$$

The objective is optimized using convex optimization techniques, with the sparsity of $S$ speeding up the process. Note that GR optimizes all $v_k$ entirely independently. As explained before, this carries a risk of overfitting: when there are more leaves than training instances, there are infinitely many ways to perfectly fit the training data. L2 regularization ($r(v) = ||v||_2^2$) is used to counter overfitting. In addition to this, GR combines neighboring leaves if the difference in their values is sufficiently small. This operation leads to a reduction of the size of the trees.

**Leaf Refinement with L1 Ensemble Pruning (LRL1):** Buschjäger & Morik (2023) proposed combining leaf refinement with "ensemble pruning", which removes entire trees with minimal performance loss. Contrary to GR, size reduction is an explicit goal of their approach.

Recall our earlier formulation of forests as $\boldsymbol{T}(x) = \sum_{i,j} w_i \mathbb{1}_{i,j}(x) \hat{y}_{i,j}$, which ultimately was simplified to $\boldsymbol{T}(x) = s(x) \cdot v$. Rather than combining the product of $w_i$ and $\hat{y}_{i,j}$ into one parameter $v_k$, LRL1 keeps them separate and tries to push the $w_i$ to 0. Thus, it keeps separate vectors $w$ and $\hat{y}$ and optimizes the following objective:

$$\min_{\hat{y},w} \frac{1}{N} \sum_{i=1}^{N} \ell(s(x_i) D_w \hat{y}, y_i) + \alpha ||w||_1 \qquad (4)$$

where $\alpha$ is the regularization parameter, and $D_w$ is a diagonal matrix storing tree coefficients $w$. This non-smooth objective is optimized using a proximal approach (Buschjäger & Morik, 2023).

An essential difference between LOP and both GR and LRL1 is that LOP does not optimize all the $v_k$ values independently, at least not for the original forest. It consecutively solves optimization problems with much fewer parameters (the $b_n$ and $c_n$ parameters) in order to prune as much as possible. It is still the case that at the deepest level, when $k$ is maximal, all remaining leaf values (the $b_n$) are optimized independently, but the number of leaves is typically dramatically reduced by the preceding pruning (more details in Appendix B). Thus, LOP solves more, but simpler, optimization problems, and is inherently less prone to overfitting.

At level 0, LOP behaves similarly to LRL1 in that it removes entire trees and it uses no bias terms at this level. It still differs in that LRL1 optimizes all leaf values, while LOP only rescales them by one scaling factor per tree. For $l > 0$, LOP's use of bias terms is essential and there is no counterpart in LRL1.

In summary, LOP can prune nodes at any level of the tree, whereas LRL1 is limited to pruning only at level 0, GR (repeatedly) merges leaves at the lowest level, and FP removes all nodes below a selected level in each tree.

*Table 1.* **Compression results on XGBoost models.** For the original models, we report their number of leaves (#Leaf) and balanced accuracy on the test set (Bacc). For the compression techniques, we report the compression ratio and drop in balanced accuracy versus the original model. A $8\times$ compression ratio means that the compressed model has 8 times fewer leaf nodes than the original ensemble. A positive (negative) balanced accuracy difference means the compressed model is less (more) accurate than the original model. The best compression ratios are given in **bold**.

| | #Leaf | Compress. ratio ($\times$) | | | | | Bacc | Diff. Bacc (% Point) | | | | |
|---|---|---|---|---|---|---|---|---|---|---|---|---|
| Dataset | XGB | GR | IC | LRL1 | FP | LOP | XGB | GR | IC | LRL1 | FP | LOP |
| Mnist | 773.0 | 6.4 | 2.8 | 1.8 | 3.2 | **8.0** | 99.0 | 0.5 | 0.4 | 0.0 | 0.4 | 0.6 |
| Electricity | 2548.4 | 2.8 | 1.5 | 1.2 | 1.5 | **5.2** | 86.2 | 0.1 | 0.4 | -0.1 | 0.2 | 0.5 |
| Jannis | 3319.9 | 11.1 | 6.6 | 1.4 | 14.7 | **18.1** | 77.0 | 0.3 | 0.4 | 0.3 | 0.3 | 0.6 |
| Vehicle | 396.2 | 2.2 | 2.6 | 1.1 | 2.0 | **8.0** | 94.4 | 0.1 | 1.0 | -0.4 | 1.1 | 1.4 |
| DryBean | 1117.8 | 4.3 | 6.7 | 1.6 | 19.6 | **28.8** | 91.2 | 0.0 | 0.3 | 0.0 | 0.2 | 0.3 |
| California | 2232.1 | 5.8 | 2.2 | 1.3 | 2.5 | **9.1** | 88.8 | 0.2 | 0.4 | -0.1 | 0.4 | 0.6 |
| Compas | 1525.3 | 26.7 | 15.4 | 2.2 | 240.8 | **356.8** | 65.3 | -0.4 | -0.2 | 0.0 | -0.5 | -0.1 |
| Volkert | 1101.0 | 8.0 | 3.9 | 2.0 | 4.8 | **8.3** | 98.7 | 0.5 | 0.4 | 0.2 | 0.4 | 0.6 |
| Adult | 2214.2 | 3.6 | 3.9 | 1.6 | **37.5** | 31.8 | 75.9 | -0.3 | 0.2 | 0.1 | 0.1 | -0.2 |
| Ijcnn1 | 2421.1 | 4.2 | 1.2 | 1.2 | 2.4 | **6.5** | 93.3 | -0.4 | 0.4 | -0.6 | 0.2 | 0.4 |
| MiniBooNE | 3239.6 | 6.0 | 2.0 | 1.3 | 2.5 | **6.2** | 92.8 | 0.5 | 0.5 | 0.2 | 0.4 | 0.6 |
| Phoneme | 1346.1 | 2.1 | 1.7 | 1.3 | 4.1 | **7.5** | 84.7 | 0.2 | 0.6 | -0.2 | 0.7 | 1.3 |
| Spambase | 792.3 | 3.0 | 2.1 | 1.5 | 3.9 | **8.5** | 94.0 | 0.4 | 0.6 | -0.1 | 0.6 | 0.9 |
| Credit | 2230.4 | 24.8 | 18.6 | 2.0 | 133.7 | **196.6** | 76.4 | 0.2 | 0.0 | 0.7 | 0.4 | 0.5 |
| average | 1804.1 | 7.9 | 5.1 | 1.5 | 33.8 | **50.0** | 87.0 | 0.1 | 0.4 | 0.0 | 0.4 | 0.6 |

## 5. Experiments

We empirically evaluate LOP and aim to answer the following questions: (Q1) Given a learned binary classification forest, what is the effect of compression on model size and performance? (Q2) How does LOP's compression affect energy consumption, memory footprint, and verifiability of models? (Q3) How sensitive is LOP to its hyperparameters $\Delta$ and $R$?

Appendix C addresses two additional questions: (Q4) What is the effect of compression on regression forests? (Q5) How does the runtime of compression scale with forest size?

### 5.1. Experimental setup

We compress XGBoost and RandomForest models because they are widely used, but LOP is equally applicable to other types of tree ensembles.

We compare LOP to the original ensembles (XGB or RF) and to four baseline methods. Individual Contribution (Lu et al., 2010) (IC) ranks and selects a subset of trees in a forest based on accuracy and diversity. It performed well in previous evaluations (Buschjäger & Morik, 2023). It represents the *coarsest extreme* by operating only at the tree level, serving as a level 0 ablation of our method. Beyond this method, we also consider the previously described closely related methods of Global Refinement (Ren et al., 2015) (GR), leaf refinement combined with L1 ensemble

pruning (Buschjäger & Morik, 2023) (LRL1), and Forest-Prune (Liu & Mazumder, 2023a) (FP).

We consider 14 binary classification benchmark datasets.[3] available on OpenML (Vanschoren et al., 2013): Compas, Vehicle, Spambase, Phoneme, Adult, Ijcnn1, Mnist (2 vs. 4), DryBean (6 vs. rest), Volkert (2 vs. 7), Credit, California, MiniBooNE, Electricity, and Jannis.

We use 5-fold cross-validation with 3 folds for training (both training the ensemble and compressing it), 1 for validation, and 1 for testing. In each fold, we train models on all combinations of the following hyperparameters:

| | **XGBoost** | **RandomForest** |
|---|---|---|
| $M$ | $\in [10, 25, 50, 100]$ | $\in [50, 100, 250]$ |
| $D$ | $\in [4, 6, 8]$ | $\in [10, 15]$ |
| $\eta$ | $\in [0.1, 0.25, 0.5, 1.0]$ | not applicable |

with $M$ the number of trees, $D$ the maximum depth of the trees and $\eta$ the learning rate in XGBoost. This yields 48 XGBoost models and 6 RandomForest models, to which we then apply the different compression algorithms.

The validation set is used to tune the regularization hyperparameter for LOP, GR, LRL1 and FP. More specifically, its optimal value is the one that leads to the smallest model and is within a maximum drop $\Delta = 0.5\%$ on the validation set's balanced accuracy. The same is done to find the optimal

---

[3]Dataset characteristics can be found in Table A1

*Table 2.* **Compression results on RandomForest models.** For the original models, we report their number of leaves (#Leaf) and balanced accuracy on the test set (Bacc). For the compression techniques, we report the compression ratio and drop in balanced accuracy versus the original model; see the caption of Table 1 for precise definitions. The best compression ratios are given in **bold**.

| | **#Leaf** | **Compress. ratio ($\times$)** | | | | | **Bacc** | **Diff. Bacc (% Point)** | | | | |
| Dataset | RF | GR | IC | LRL1 | FP | LOP | RF | GR | IC | LRL1 | FP | LOP |
|---|---|---|---|---|---|---|---|---|---|---|---|---|
| Mnist | 20039.0 | 61.9 | 22.7 | 2.4 | 22.5 | **74.1** | 99.3 | 0.5 | 0.4 | 0.3 | 0.6 | 0.8 |
| Electricity | 135271.6 | 114.4 | 21.9 | 7.8 | 19.5 | **125.0** | 85.3 | 0.2 | 0.3 | -0.6 | 0.6 | 0.5 |
| Jannis | 183003.7 | 158.8 | 6.5 | 2.1 | 3.7 | **249.0** | 77.5 | 0.5 | 0.5 | 0.1 | 0.3 | 0.5 |
| Vehicle | 5111.0 | 7.7 | 16.9 | 1.0 | 12.8 | **46.4** | 94.9 | -0.1 | 1.2 | 0.5 | 0.7 | 1.3 |
| DryBean | 26042.8 | 16.7 | 16.9 | 1.2 | **202.0** | 66.6 | 91.5 | 0.4 | 0.4 | 0.1 | 0.6 | 0.9 |
| California | 89481.1 | 174.4 | 22.5 | 1.7 | 13.0 | **196.2** | 88.4 | 0.4 | 0.6 | -0.1 | 0.5 | 0.5 |
| Compas | 68893.7 | 473.7 | 15.3 | 1.0 | 8998.6 | **12086.8** | 63.3 | -1.4 | 0.5 | 0.3 | -0.7 | -1.3 |
| Volkert | 25773.5 | 90.3 | 31.5 | 9.0 | 38.1 | **110.9** | 98.4 | 0.5 | 0.4 | 0.3 | 0.5 | 0.6 |
| Adult | 99148.2 | 194.6 | 90.9 | 11.0 | 322.6 | **1046.8** | 75.2 | -0.6 | -0.3 | -0.4 | 0.1 | 0.0 |
| Ijcnn1 | 131413.7 | 199.6 | 80.0 | 3.0 | 59.0 | **227.9** | 89.8 | -0.1 | -0.2 | -4.2 | -0.6 | 0.4 |
| MiniBooNE | 149448.7 | **132.5** | 14.1 | 5.0 | 8.5 | 121.4 | 92.6 | 0.5 | 0.6 | -0.1 | 0.5 | 0.7 |
| Phoneme | 31578.2 | 8.0 | 12.5 | 2.0 | 15.9 | **23.7** | 87.0 | 0.3 | 0.8 | 0.0 | 1.1 | 2.2 |
| Spambase | 18634.6 | 44.5 | 23.5 | 1.3 | 32.5 | **106.7** | 93.4 | 0.1 | 0.9 | 0.0 | 1.1 | 0.8 |
| Credit | 91428.9 | 216.9 | 9.1 | 1.0 | 26.9 | **1859.5** | 77.5 | 0.3 | 0.7 | 0.0 | 0.5 | 0.7 |
| average | 76804.9 | 135.3 | 27.5 | 3.5 | 698.3 | **1167.2** | 86.7 | 0.1 | 0.5 | -0.3 | 0.4 | 0.6 |

number of trees in IC. Additionally, we set $R = 2$ for LOP.

For GR and LOP, we use scikit-learn (Pedregosa et al., 2011). GR solves its optimization problem using a linear SVM with L2 regularization. LOP[4] uses logistic regression and optimizes the negative log-likelihood with L1 regularization. For LRL1[5] and FP[6], we adapt their public implementations to work with our internal representation of tree ensembles.

All experiments are run on an Intel(R) Core(TM) i7-12700 with 64GB of memory. Each individual compression task is run on a single thread (i.e., timing results measure single-thread performance). We apply a time limit of 6 hours for compressing a single model.[7]

### 5.2. Results

**Q1: Comparing compression, predictive performance, and runtime.** The left hand side of Tables 1 and 2 show the compression factors obtained by each method averaging over all learned XGBoost and RandomForest models per dataset. For both XGBoost and RandomForest, LOP systematically achieves the best compression factor. Its average compression factors range from 5 to 356 on XGBoost models and from 23 to 12 086 on RandomForest models. LOP typically has a 2-3x better compression factor than its

---

[4] https://github.com/ML-KULeuven/lop_compress
[5] https://github.com/sbuschjaeger/leaf-refinement-experiments
[6] https://github.com/mazumder-lab/ForestPrune
[7] The time limit was only exceeded for a small number of Random-Forest models; see Table A8 in the Appendix for an overview.

nearest competitor and can be up to 3 orders of magnitude better than the worst competitor. LOP benefits from having much more fine-grained control of where it prunes than its competitors. Moreover, it can perform multiple rounds of pruning unlike FP, IC, and LRL11. However, the sensitivity analysis discussed in Q3 indicates that even one round of LOP yields substantially more compression than its competitors. FP and GR typically achieve the next best compression, though which of the two is best varies: each outperforms the other on half the datasets. On XGBoost when FP outperforms GR, it tends to have much bigger compression factors whereas this trend does not hold on RandomForest. IC typically is the fourth best approach with LRL1 uniformly achieving the least amount of compression. Generally, all methods achieve better compression on RandomForest models, likely because these models are (much) bigger than the XGBoost ones.

The right-hand side of Tables 1 and 2 shows the difference in balanced accuracy between the compressed ensemble and the original ensemble for respectively the XGBoost and RandomForest models. All compression methods result in models that, on average, only have a small loss. The difference in performance is typically less than 0.5 percentage points. For LOP, we see that the accuracy loss on the test set is quite close to $0.5\%$ (i.e., the chosen value for $\Delta$), which suggests that no overfitting occurs on the validation set. Even with its far superior compression rates, LOP maintains balanced accuracy as well as the other methods.

The above tables show compression rates and balanced ac-

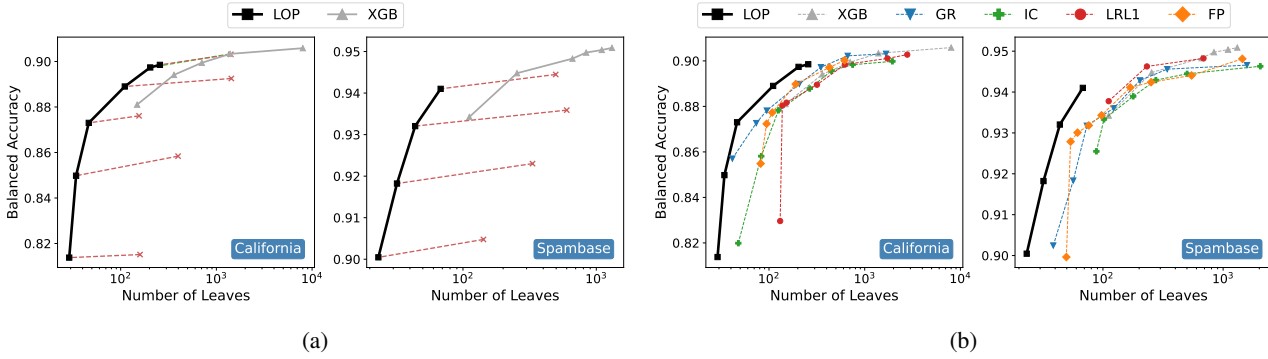

*Figure 2.* Comparison of Pareto fronts for (a) LOP vs. XGBoost (b) and all methods on two representative datasets. The x-axis shows the number of leaves in log scale and the y-axis shows the balanced accuracy. In (a), the dashed line connect a compressed LOP model to the performance of its original XGBoost model. These are colored red if the original model is not on the Pareto front for XGBoost.

curacies separately, but a user may want to select the "best" model according to some size versus performance trade-off mechanism. This can be visualized using a Pareto curve where a model appears on the curve only if achieving a better balanced accuracy requires a larger ensemble or making a smaller ensemble requires sacrificing balanced accuracy.

A question that comes to mind is: can we preselect models to compress based on the Pareto front? Figure 2a shows that the answer is no: Pareto-optimal compressed models can be obtained from Pareto-suboptimal XGBoost models.[8]

Figure 2b shows the Pareto fronts obtained for the different compression methods applied to the XGBoost models on two datasets.[9] Again LOP clearly outperforms the other methods, in terms of how much size reduction can be obtained for a given level of balanced accuracy.

Finally, Figure A8 and A9 in the Appendix show the runtimes of each compression method on the XGBoost and RandomForest models, respectively. IC is the fastest method, typically taking on the order of seconds, but it yields very little compression. LOP and FP exhibit similar runtimes. GR is slightly slower than LOP and FP on the XGBoost models, but much slower on the RandomForest models that are generally larger in size. LRL1 is clearly the slowest approach.

**Q2: Robustness checking and resource usage.** Next, we evaluate the effect of compression on two use cases where it is beneficial to have smaller models: robustness checking and resource-constrained devices. Empirical robustness measures the average distance to the nearest adversarial example for a given set of correctly classified normal examples, which is a NP-complete problem for decision forests (Kantchelian et al., 2016). It is often used as a measure of a model's susceptibility to evasion at-

tacks (Moshkovitz et al., 2021; Devos et al., 2021a). For each fold, we use Kantchelian et al.'s (2016) exact MILP approach to find the nearest adversarial example for 500 randomly selected test examples using a global timeout of 30 minutes.

Table 3 shows that the median time for finding the nearest adversarial example on an XGBoost model compressed by LOP is 2.5 to 10 times less than for models compressed by its competitors. It is much more efficient to perform robustness checking on the compressed models than the original XGBoost models. Interestingly, Table A7 in the appendix shows that LOP returns more robust ensembles.

For resource constrained devices, two things are relevant. First, prediction time is correlated with energy (i.e., battery) use (Verachtert et al., 2016). Second, the memory footprint of the model is also important. However, both factors depend (strongly) on the precise implementation of a decision forest and therefore, we look at two proxies instead. Because test time is typically dominated by the number of splits to evaluate, we report the median number of nodes that an instance passes through when using the ensemble to make a prediction. For memory footprint, we use a formula from prior work that assumes that storing each node requires $17 + 4 \cdot C$ bytes, with $C$ the number of classes (Buschjäger & Morik, 2023). Table 3 shows that LOP leads to better ensembles on both metrics. In particular, LOP produces models that use substantially less memory than the other approaches.

**Q3: Sensitivity analysis of LOP.** We now explore the effect of two of LOP's hyperparameters: (1) $\Delta$ which controls the maximum allowable loss in predictive performance compared to the original ensemble, and (2) $R$ which specifies the number of rounds of level-by-level compression that is performed. We now investigate how varying these hyperparameters affects performance on four representative datasets: Adult, California, Phoneme, and Spambase.

---

[8]Figure A2 in the appendix shows these for all datasets. The same is true for other baselines, see Figures A3, A4, A5 and A6

[9]Figure A7 in the appendix shows these for all datasets.

*Table 3.* Distribution (25%, Median and 75%) of robustness checking runtime (in seconds), number of splits to evaluate at test time and model size (in KB) before and after applying the different compression methods to an XGBoost ensemble. Results are obtained by (1) finding the median per XGBoost parameter combination (i.e., number of trees, max depth and learning rate) over the datasets and folds and (2) computing the quantiles over these 48 median values (i.e., the number of parameter combinations).

|  | **Verif. time (s)** | | | **#Splits to evaluate** | | | **Memory footprint (KB)** | | |
|---|---|---|---|---|---|---|---|---|---|
|  | 25% | Median | 75% | 25% | Median | 75% | 25% | Median | 75% |
| XGB | 116.49 | 369.24 | 1092.64 | 94.5 | 197.00 | 386.13 | 45.6 | 92.3 | 211.0 |
| GR | 22.91 | 44.31 | 78.39 | 40.88 | 90.75 | 151.00 | 11.5 | 22.4 | 33.0 |
| IC | 31.90 | 70.75 | 134.06 | 37.00 | 66.25 | 120.00 | 17.1 | 28.6 | 48.4 |
| LRL1 | 65.30 | 148.28 | 447.86 | 71.88 | 147.25 | 226.50 | 29.6 | 58.8 | 130.0 |
| FP | 17.27 | 32.05 | 79.67 | 36.75 | 53.75 | 101.50 | 9.7 | 15.3 | 29.2 |
| LOP | **9.60** | **13.34** | **21.17** | **21.13** | **28.75** | **35.75** | **5.6** | **8.2** | **12.0** |

*Table 4.* Effect of varying $\Delta \in \{0.25, 0.5, 1, 2\}$ and $R \in \{1, 2, 3\}$ while keeping, respectively, $R = 2$ and $\Delta = 0.5$ on LOP's performances on four representative datasets. The left hand side shows compression ratios while the right hand side shows the difference in balanced accuracy (i.e., percentage point difference) between LOP's compressed model and the original ensemble. For each dataset, results are averaged over all 48 learned XGBoost ensembles and corresponding compressed models obtained by LOP.

|  | **Compress. ratio ($\times$)** | | | | | | | **Diff. Bacc (% Point)** | | | | | | |
|---|---|---|---|---|---|---|---|---|---|---|---|---|---|---|
|  | $R=2$ | | | | $\Delta=0.5$ | | | $R=2$ | | | | $\Delta=0.5$ | | |
| Dataset | $\Delta=0.25$ | $=0.5$ | $=1$ | $=2$ | $R=1$ | $=2$ | $=3$ | $\Delta=0.25$ | $=0.5$ | $=1$ | $=2$ | $R=1$ | $=2$ | $=3$ |
| Adult | 26.0 | 31.8 | 45.3 | 67.5 | 23.0 | 31.8 | 35.5 | -0.4 | -0.2 | 0.5 | 1.4 | 0.1 | -0.2 | -0.1 |
| California | 7.3 | 9.2 | 11.9 | 17.4 | 7.0 | 9.2 | 10.1 | 0.3 | 0.6 | 1.1 | 2.0 | 0.5 | 0.6 | 0.6 |
| Phoneme | 6.7 | 7.5 | 10.4 | 6.0 | 6.0 | 7.5 | 8.6 | 1.2 | 1.3 | 1.8 | 2.6 | 1.0 | 1.3 | 1.4 |
| Spambase | 7.0 | 8.5 | 12.1 | 17.2 | 6.8 | 8.5 | 9.2 | 0.7 | 0.9 | 1.4 | 2.4 | 0.8 | 0.9 | 0.9 |

Table 4 shows how varying $\Delta \in \{0.25, 0.5, 1, 2\}$ while keeping $R = 2$ affects both the compression ratio and the difference in balanced accuracy obtained by LOP compared to the base XGBoost ensembles. There is a clear effect: increasing $\Delta$ yields more compression but harms predictive performance. Setting $\Delta = 0.25$ yields slightly less compression than $\Delta = 0.5$ but does give better predictive performance. Moving from $\Delta = 0.5$ to $\Delta = 2$ achieves twice as much compression but tends to lead to slightly bigger drops in predictive performance.

Alternatively, Table 4 shows how varying $R \in \{1, 2, 3\}$ while keeping $\Delta = 0.5$ affects the compression ratio and predictive performance obtained by LOP. Going from $R = 1$ to $R = 2$, yields larger wins in compression while slightly degrading performance on three datasets but marginally improving it on the Adult dataset. Performing a third round essentially does not change predictive performance while still allowing for (slightly) more compression. This increases runtime (see Figure A11 in the Appendix).

## 6. Conclusion

We propose LOP, a novel level-by-level approach to compressing previously learned tree ensembles. Empirically, LOP achieves compression ratios that are an order of magni-

tude larger than those of prior methodologies while obtaining similar predictive performance to the baselines. Moreover, we show that LOP has several other benefits. In terms of robustness checking, LOP produces more robust models than its competitors and it is faster to check the robustness of LOP models. For resource constrained devices, LOP offers more efficiency in terms of making predictions and produces models that have a much smaller memory footprint.

## Acknowledgements

This research is supported by the Flemish government under the "Onderzoeksprogramma Artificiële Intelligentie (AI) Vlaanderen" programme (HB, JD & WM), the European Union's Horizon Europe Research and Innovation program under the grant agreement TUPLES No. 101070149 (LD & JD), the KU Leuven Research Fund (C2E/23/007, HB & WM) and iBOF/21/075 (TM & JD).

## Impact Statement

This paper presents work whose goal is to advance the field of Machine Learning, and the standard societal consequences of such work apply. More particularly, the work positively affects sustainability and reliability (fairness, robustness) of machine-learned models.

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

## A. Usefulness of leaf refinement

We claim in the paper that individual trees may need to be suboptimal in order for an ensemble to be optimal. We now make that claim more precise.

Trees induce a partition of the input space through their leaves. Given a tree $T$, let $s(x) \in \{1, 2, \ldots, n\}$ be the index of the leaf $x$ is sorted into, and let $\mathcal{X}_i = \{x \in \mathcal{X} | s(x) = i\}$. The tree defines a piecewise constant function that is constant within each $\mathcal{X}_i$. From a least squares point if view, the tree is optimal if its prediction in leaf $i$ is the mean of all target values of instances in $\mathcal{X}_i$.

An ensemble similarly partitions the input space into subsets $\mathcal{X}_k$, $k \in \{1, 2, \ldots L\}$ (with $k$ indexing all leaves in the ensemble), such that the ensemble's predictions are constant within each subset. The ensemble is optimal if its predictions in each subset equals the mean of all target values in that subset.

Our claim, now, is that ensembles exist such that if all individual trees $T_i$ in an ensemble are optimal in the above sense, the ensemble is not optimal, and vice versa. To prove the claim, it suffices to show an example.

Figure A1 show two decision stumps (trees of depth 1) with one-dimensional inputs. Each tree represents a piecewise constant function; the optimal value in each leaf is the mean of the target function in the interval covered by the leaf. The forest is then a piecewise constant function with three intervals, but the constants are biased: they differ from the mean of the target function over the interval. The bias can be removed by changing the leaf values, which then leads to trees that are not individually optimal.

This examples shows how leaf values that are optimal in an individual tree are suboptimal in the forest, and vice versa. This is an argument for optimizing $v$ after the whole forest has been learned.

## B. Size of optimization problems

Consider a forest with $M$ trees, and consider the optimization problems of the form discussed in the paper, which contain a $c_n$ and $b_n$ parameter for each active node $n$ (except root nodes, which have only $c_n$, and leaf nodes, which have only $b_n$).

The number of parameters to optimize at level 0 is $M$ (and not $2M$) because the $b$'s are excluded on this level. If the pruning results in $M'$ remaining trees, then on level 1 the number of parameters is $2 \cdot 2 \cdot M'$ (2 parameters for each of the $2M'$ active nodes on level 1), with $M' \leq M$. On each level, the algorithm tries to prune, and each pruned subtree becomes a leaf $n$ that contributes only 1 parameter $b_n$ to all lower levels, rather than $2, 4, 8, \ldots$ parameters as we go

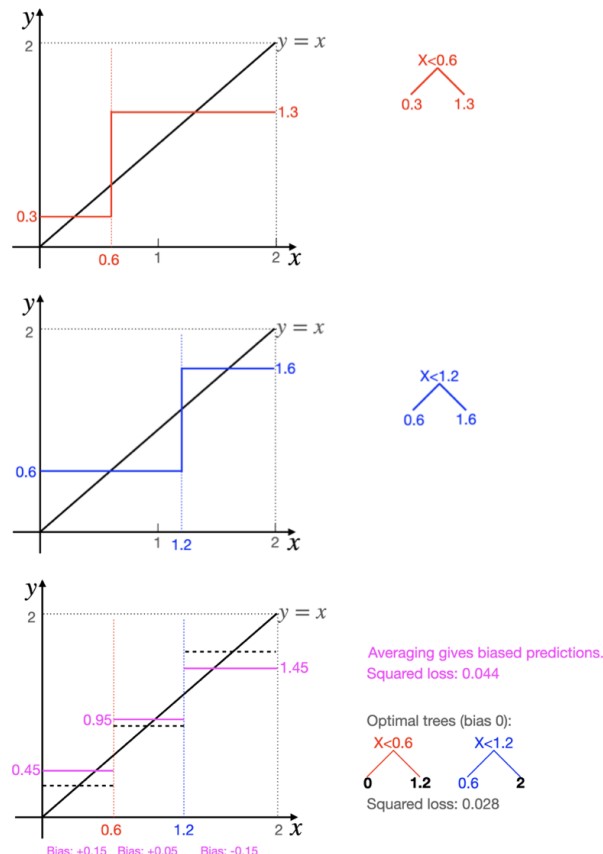

*Figure A1.* Bias in forests. Consider approximating $y = x$ with decision stumps with random splits on $x$. The first stump (red) splits on $x < 0.6$, the second (blue) on $x < 1.2$. Both have in their leaves an optimal prediction (the population mean of all instances in that leaf). A forest that averages their predictions represents a partitioning of the input space into 3 intervals, with predictions showing a clear bias w.r.t. the optimal prediction for each interval (dashed line). An unbiased forest exists with exactly the same splits: it suffices to store different values in the leaves, which are then suboptimal from an individual tree's point of view.

*Table A1.* Properties of the binary classification datasets: Name, number of examples, number of features, and class prior $\alpha$ (i.e., proportion of positive examples).

| Name | # Examples | # Features | $\alpha$ |
|---|---|---|---|
| Mnist | 13814 | 784 | 0.49 |
| Electricity | 38474 | 8 | 0.50 |
| Jannis | 57580 | 54 | 0.50 |
| Vehicle | 846 | 18 | 0.49 |
| DryBean | 13611 | 16 | 0.19 |
| California | 20634 | 8 | 0.50 |
| Compas | 4966 | 11 | 0.50 |
| Volkert | 24325 | 180 | 0.53 |
| Adult | 48842 | 32 | 0.24 |
| Ijcnn1 | 141691 | 22 | 0.10 |
| MiniBooNE | 72998 | 50 | 0.50 |
| Phoneme | 5404 | 5 | 0.71 |
| Spambase | 4601 | 57 | 0.39 |
| Credit | 16714 | 10 | 0.50 |

*Table A2.* Properties of the regression datasets: Name, number of examples and number of features.

| Name | # Examples | # Features |
|---|---|---|
| Abalone | 4177 | 8 |
| Ailerons | 13750 | 33 |
| Cpu | 8192 | 12 |
| Houses | 20640 | 8 |
| House16H | 22784 | 16 |
| WineQuality | 6497 | 11 |
| Elevators | 16599 | 16 |

deeper. Generally, at level $l$, if there are $a$ active nodes, of which $a_1$ are leaves-on-a-higher-level and $a_2$ are internal nodes, there are $a_1 + 2 * a_2 \leq 2a$ parameters. At the lowest level $d$, this number can be $2 \cdot 2^d \cdot M$ in the worst case, if nothing ever gets pruned; but the point is of course that LOP does prune a lot, starting at the upper levels.

## C. Detailed experimental results

In this section, we provide more detailed results about the three experimental questions in the main text. Finally, we answer the additional two questions: (Q4) What is the effect of compression on regression forests? (Q5) How does the runtime of compression scale with forest size?

**Q1: Comparing compression, predictive performance and runtime.** Tables 1 and 2 in the main text show the compression factors obtained by the considered compression algorithms, averaged over respectively all learned XGBoost and RandomForest settings per dataset. The main body of the paper focused on discussing LOP's performance rela-

tive to its competitors. Here, we detail the differences in compression between the baselines. While worse than LOP, FP and GR offer the next best performance. On XGBoost models, FP achieves better compression than GR half the time. However, when FP wins, it tends to result in much bigger compression factors and hence it has a higher average compression ratio than GR. For RandomForest, GR performs better on nine datasets. In contrast to the XGBoost results, sometimes GR can achieve orders of magnitude better compression than FP. While they achieve substantially less compression than LOP, GR and FP yield better compression than LRL1 and IC on both XGBoost and RandomForest models. This is because they have more fine-grained pruning mechanisms. FP can prune at any level of a tree, whereas LRL1 and IC only remove trees. GR typically applies multiple rounds of refinement/pruning until convergence, whereas LRL1 and IC only apply one round of refinement.

We now discuss the effect of compression on the number of trees retained in the ensemble. We compute the tree reduction ratio, which is the number of trees in the original ensemble divided by the number of trees in the compressed model. Tables A3 and A4 show this ratio for each method on XGBoost and RandomForest models, respectively. Higher numbers indicate fewer trees in the compressed model. On XGBoost models, ensembles compressed by LOP have the fewest number of trees. Among the baselines, IC is more effective at pruning full trees than LRL1 as it reduces the number of trees in the original forest by a factor 23.3 compared to a factor 2. Note that GR never removes trees from the ensembles. In contrast, on RandomForest models, both IC and FP reduce the number of trees more than LOP. However, the fact that LOP has better compression ratios than FP, suggests that FP's compression ratios are mainly obtained through pruning full trees.

Table 1 shows the average performance over a wide variety of XGBoost settings. However, often a user wants to understand the trade-off between model size and predictive performance. This can be visualized using Pareto curves, which show all models that are not dominated by another model. That is, a model appears on the curve only if achieving better balanced accuracy requires a larger ensemble or making a smaller ensemble requires sacrificing balanced accuracy.

Figures A2- A6 show the Pareto front for each compression algorithm versus XGBoost on all datasets. Moreover, for the compressed models on the front, they show the performance of its original, uncompressed XGBoost variant using a dashed line. Interestingly, there exists no direct mapping between a compression algorithm's Pareto front and XGBoost's Pareto front. That is, the best compressed models do not necessarily come from the best XGBoost models.

Figure A7 shows that typically LOP's Pareto front dominates

*Table A3.* The average tree reduction ratio for each dataset for the XGBoost models on the binary classification experiments. This ratio is computed as the number of trees in the original ensemble divided by the number of trees in the compressed model. Higher ratios are better. The best ratios are indicated in bold.

| | GR | IC | LRL1 | FP | LOP |
|---|---|---|---|---|---|
| Mnist | 1.0 | 5.4 | 2.5 | 2.7 | **8.1** |
| Electricity | 1.0 | 1.6 | 1.3 | 1.3 | **2.2** |
| Jannis | 1.0 | **8.5** | 1.5 | 3.7 | 8.1 |
| Vehicle | 1.0 | 4.4 | 1.2 | 2.7 | **7.5** |
| DryBean | 1.0 | 10.3 | 1.7 | 5.0 | **24.1** |
| California | 1.0 | 2.5 | 1.4 | 1.9 | **3.3** |
| Compas | 1.0 | 25.1 | 2.8 | 22.5 | **35.6** |
| Volkert | 1.0 | 7.1 | 2.5 | 2.9 | **8.2** |
| Adult | 1.0 | 4.4 | 1.6 | 4.2 | **10.3** |
| Ijcnn1 | 1.0 | 1.3 | 1.3 | 1.6 | **3.1** |
| MiniBooNE | 1.0 | 2.2 | 1.3 | 1.6 | **2.4** |
| Phoneme | 1.0 | 2.2 | 1.4 | 2.4 | **5.7** |
| Spambase | 1.0 | 3.0 | 1.6 | 2.9 | **5.9** |
| Credit | 1.0 | 23.3 | 2.0 | 10.5 | **26.8** |
| average | 1.0 | 7.2 | 1.7 | 4.7 | **10.8** |

*Table A4.* The average tree reduction ratio for each dataset for the RandomForest models on the binary classification experiments. This ratio is computed as the number of trees in the original ensemble divided by the number of trees in the compressed model. Higher ratios are better. The best ratios are indicated in bold.

| | GR | IC | LRL1 | FP | LOP |
|---|---|---|---|---|---|
| Mnist | 1.0 | **23.2** | 2.4 | 20.7 | 15.1 |
| Electricity | 1.0 | **23.1** | 8.2 | 18.8 | 20.4 |
| Jannis | 1.0 | **7.1** | 2.2 | 3.7 | 5.5 |
| Vehicle | 1.0 | **16.5** | 1.0 | 11.0 | 14.2 |
| DryBean | 1.0 | 18.6 | 1.2 | **33.3** | 14.4 |
| California | 1.0 | **22.3** | 1.8 | 12.7 | 16.5 |
| Compas | 1.0 | 15.5 | 1.0 | **80.1** | 46.9 |
| Volkert | 1.0 | **33.5** | 9.3 | 29.6 | 20.6 |
| Adult | 1.0 | **96.6** | 12.1 | 47.8 | 17.4 |
| Ijcnn1 | 1.0 | **77.9** | 3.0 | 49.3 | 29.7 |
| MiniBooNE | 1.0 | **14.8** | 5.1 | 8.4 | 9.2 |
| Phoneme | 1.0 | 13.3 | 2.0 | **14.3** | 6.8 |
| Spambase | 1.0 | 24.3 | 1.3 | **29.6** | 21.7 |
| Credit | 1.0 | 9.8 | 1.0 | 8.0 | **11.5** |
| average | 1.0 | **28.3** | 3.7 | 26.2 | 17.8 |

large parts of the other fronts. This conclusion aligns with what is found in Table 1.

Figures A8 and A9 show the distribution of compression time (s) for each method on every dataset for XGBoost models and RandomForest models, respectively. On XGBoost models, LOP takes on average 241s to compress a forest. This is equally as fast as GR, which takes 240s. FP and IC are faster, taking respectively 84s and 6s. LRL1 is clearly the slowest, taking 812s on average. On RandomForest models, which are larger than the XGBoost models, LOP takes 434s. FP and IC remain faster at respectively 250s and 17s, but they result in less compression. GR is now much slower, taking 3404s. LRL1 remains the slowest method, taking 4470s on average.

**Q2: Robustness checking and resource usage.** In addition to the time needed to compute the empirical robustness (shown in Table 3 and discussed in the main paper), it is also relevant to know how compression affects the robustness of the learned XGBoost ensembles. Table A7 reports the average test set empirical robustness for all considered approaches for each dataset. Higher values indicate a more robust model. Interestingly, applying LOP uniformly leads to models that are more robust than the original XGBoost models. More generally, LOP has the best average empirical robustness scores on nine out of 14 datasets. While IC and FP also improve the robustness of the XGBoost models, GR and LRL1 do not always increase robustness. In fact, they can actually yield less robust models than XGBoost in some cases.

**Q3: Sensitivity analysis.** Here we explore how sensitive LOP is to two of its hyperparameters: $\Delta$, which controls the maximum allowed loss in predictive performance when tuning the regularization parameter, and $R$, which denotes the number of rounds of level-by-level compression that is performed. In addition to the compression and predictive performance results shown Table 4, we now discuss how these hyperparameters affect LOP's runtime.

Figure A10 shows a boxplot of LOP's runtime for each value of $\Delta$ per dataset. Increasing $\Delta$ leads to decreased runtimes. Essentially, allowing for a larger loss in accuracy allows LOP to be more aggressive in its compression.

Figure A11 shows a boxplot of LOP's runtime for each value of $R$ per dataset. Increasing $R$ leads to longer runtimes. However, the last rounds are typically faster than the first rounds. This makes sense: The initial rounds compress the model. Hence, the optimization problem becomes smaller, leading to faster runtimes in later rounds.

**Q4: Performance on regression forests.** We now consider compressing regression forests. We omit IC because it is tailored to classification problems, and it is non-trivial to adapt it to handle regression tasks. We evaluate the methods on seven regression datasets, which are detailed in Table A2. We use 5-fold cross-validation with 3 folds for training, 1 for validation, and 1 for testing. In each fold, we train models

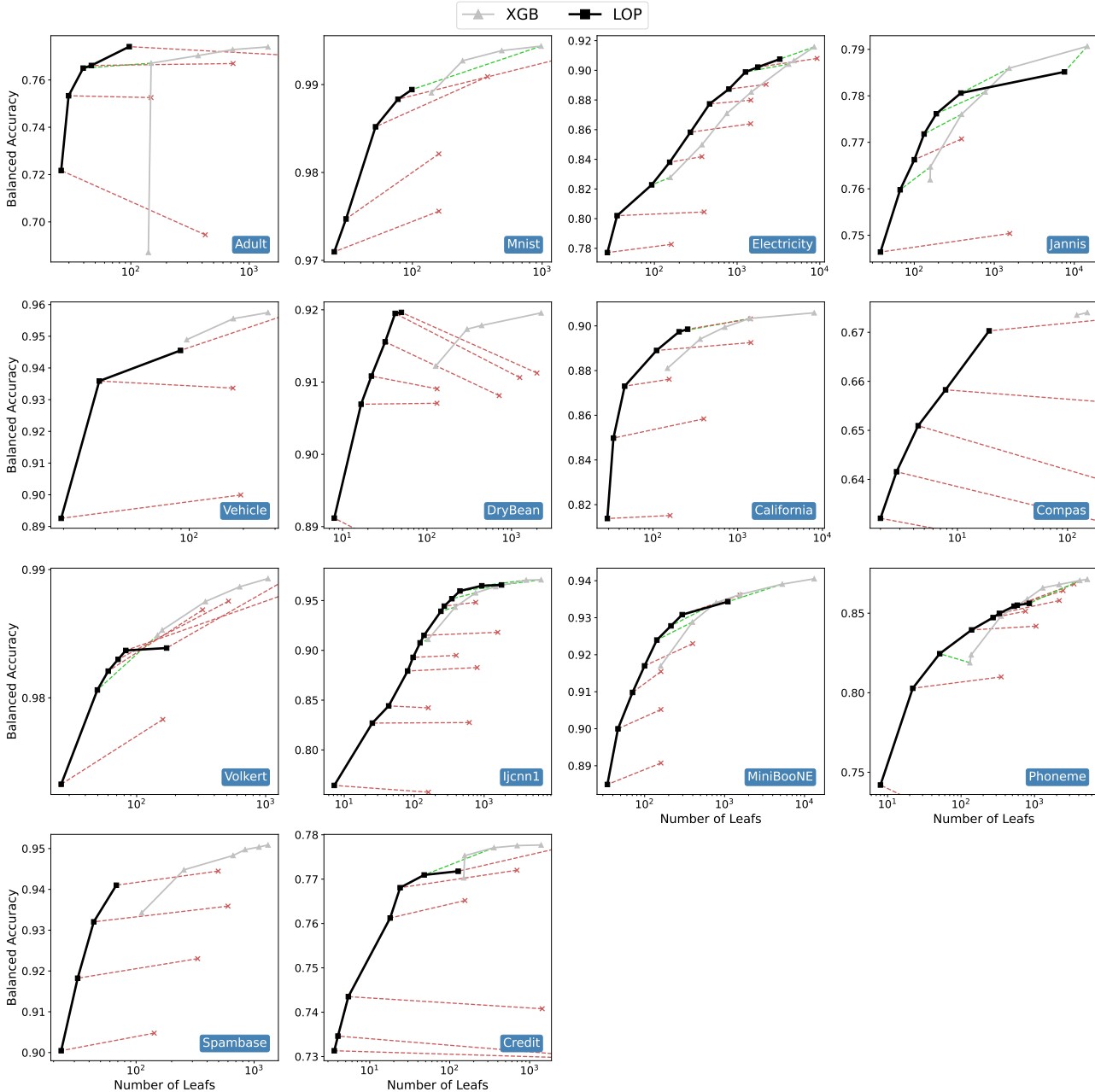

*Figure A2.* Comparison of Pareto fronts of LOP vs. XGBoost on all datasets. The x-axis shows the number of leaves in log scale and the y-axis shows the balanced accuracy. The dashed lines connect a compressed LOP model to the performance of an original XGBoost model. These are colored red if the original model is not on the Pareto front for XGBoost.

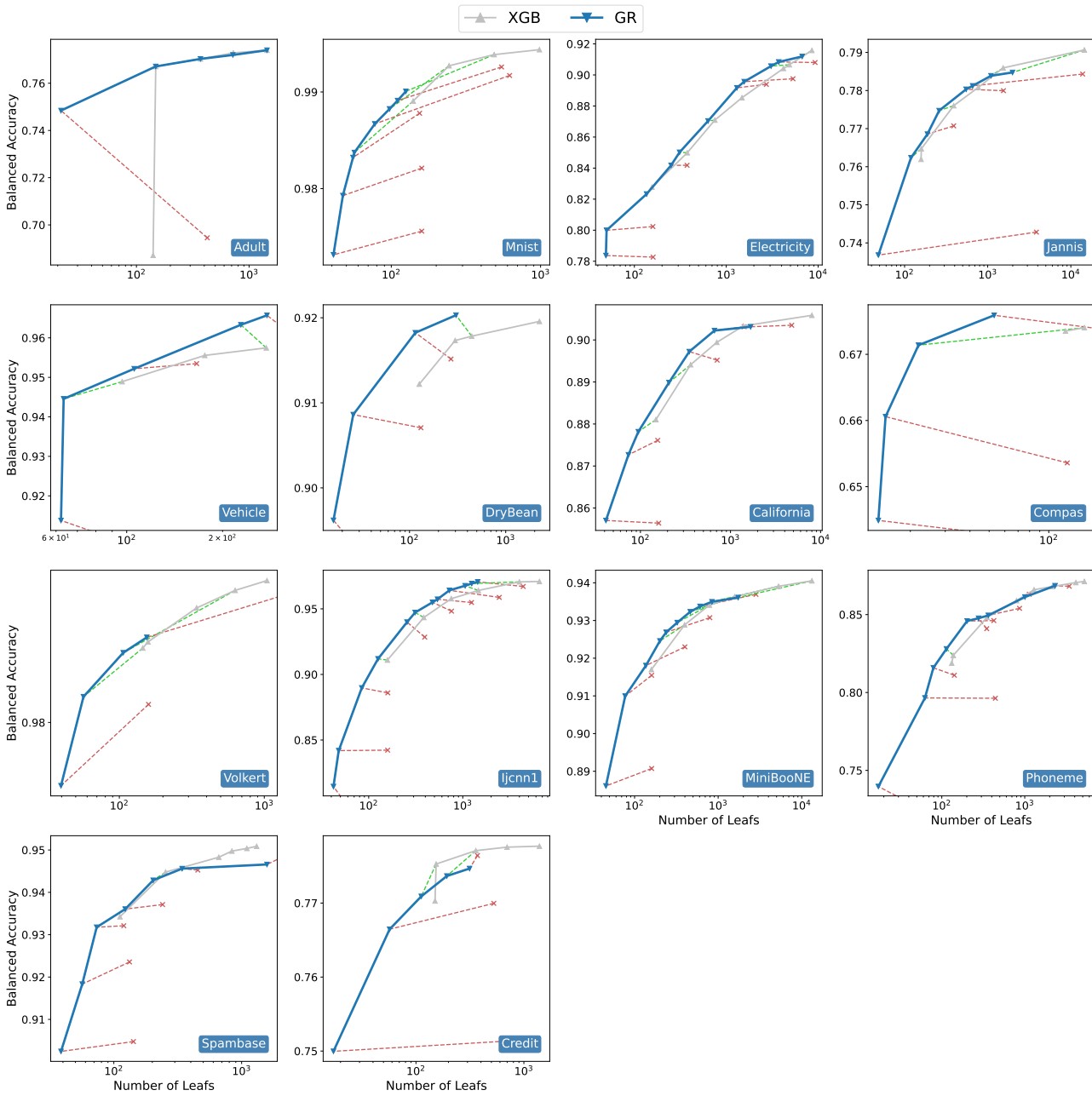

*Figure A3.* Comparison of Pareto fronts of GR vs. XGBoost on all datasets. The x-axis shows the number of leaves in log scale and the y-axis shows the balanced accuracy. The dashed lines connect a compressed GR model to the performance of an original XGBoost model. These are colored red if the original model is not on the Pareto front for XGBoost.

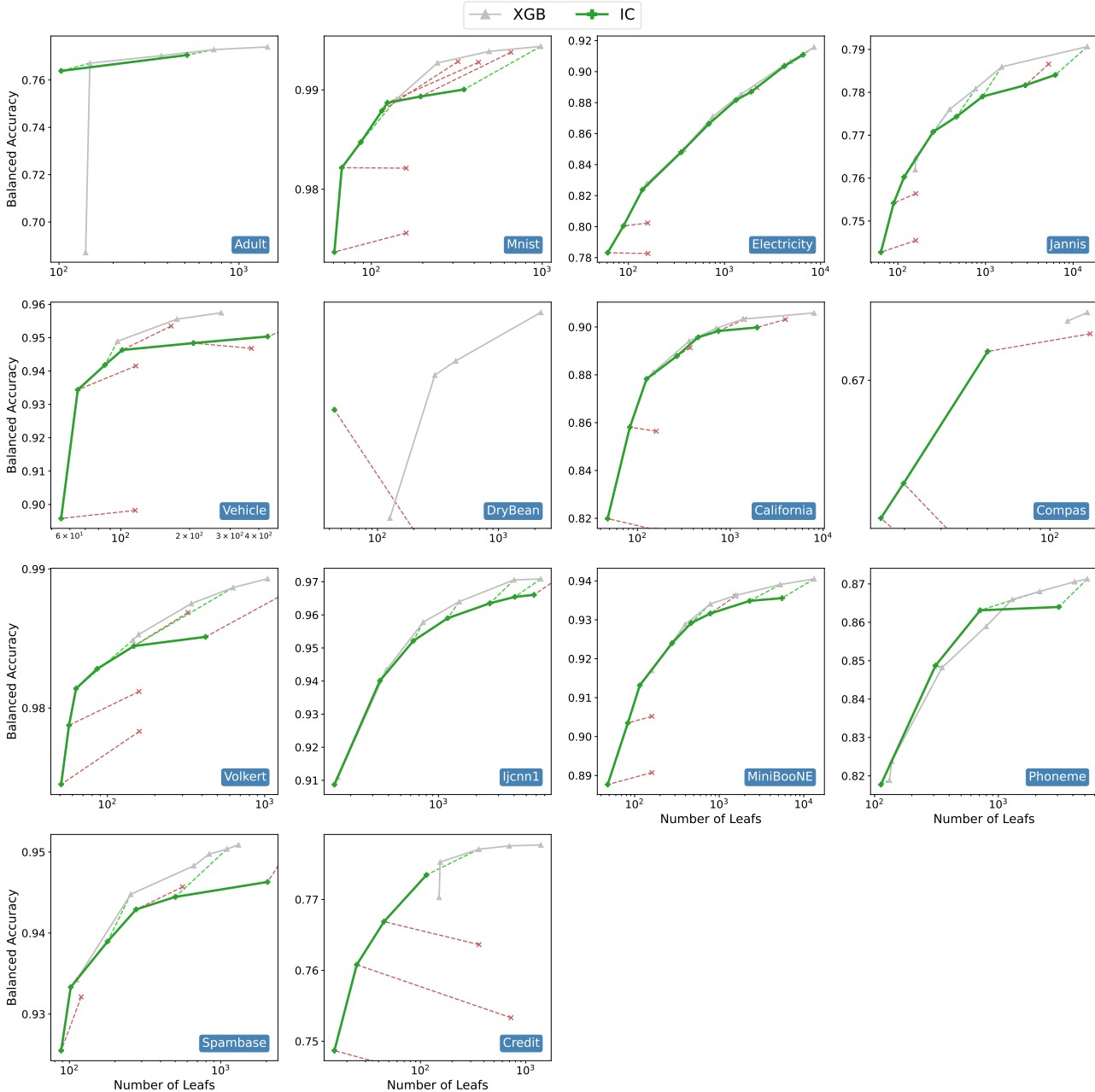

*Figure A4.* Comparison of Pareto fronts of IC vs. XGBoost on all datasets. The x-axis shows the number of leaves in log scale and the y-axis shows the balanced accuracy. The dashed lines connect a compressed IC model to the performance of an original XGBoost model. These are colored red if the original model is not on the Pareto front for XGBoost.

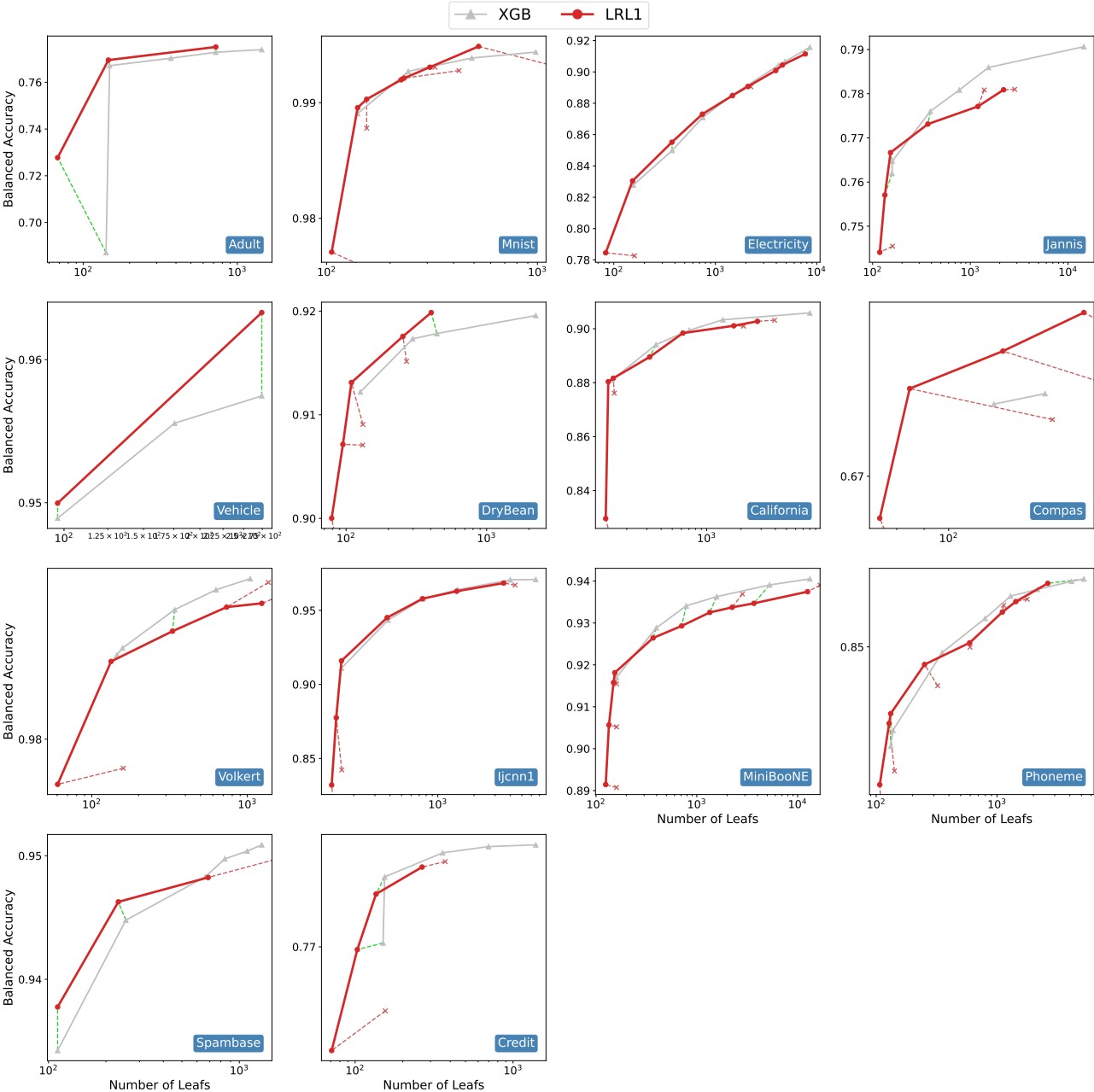

*Figure A5.* Comparison of Pareto fronts of LRL1 vs. XGBoost on all datasets. The x-axis shows the number of leaves in log scale and the y-axis shows the balanced accuracy. The dashed lines connect a compressed LRL1 model to the performance of an original XGBoost model. These are colored red if the original model is not on the Pareto front for XGBoost.

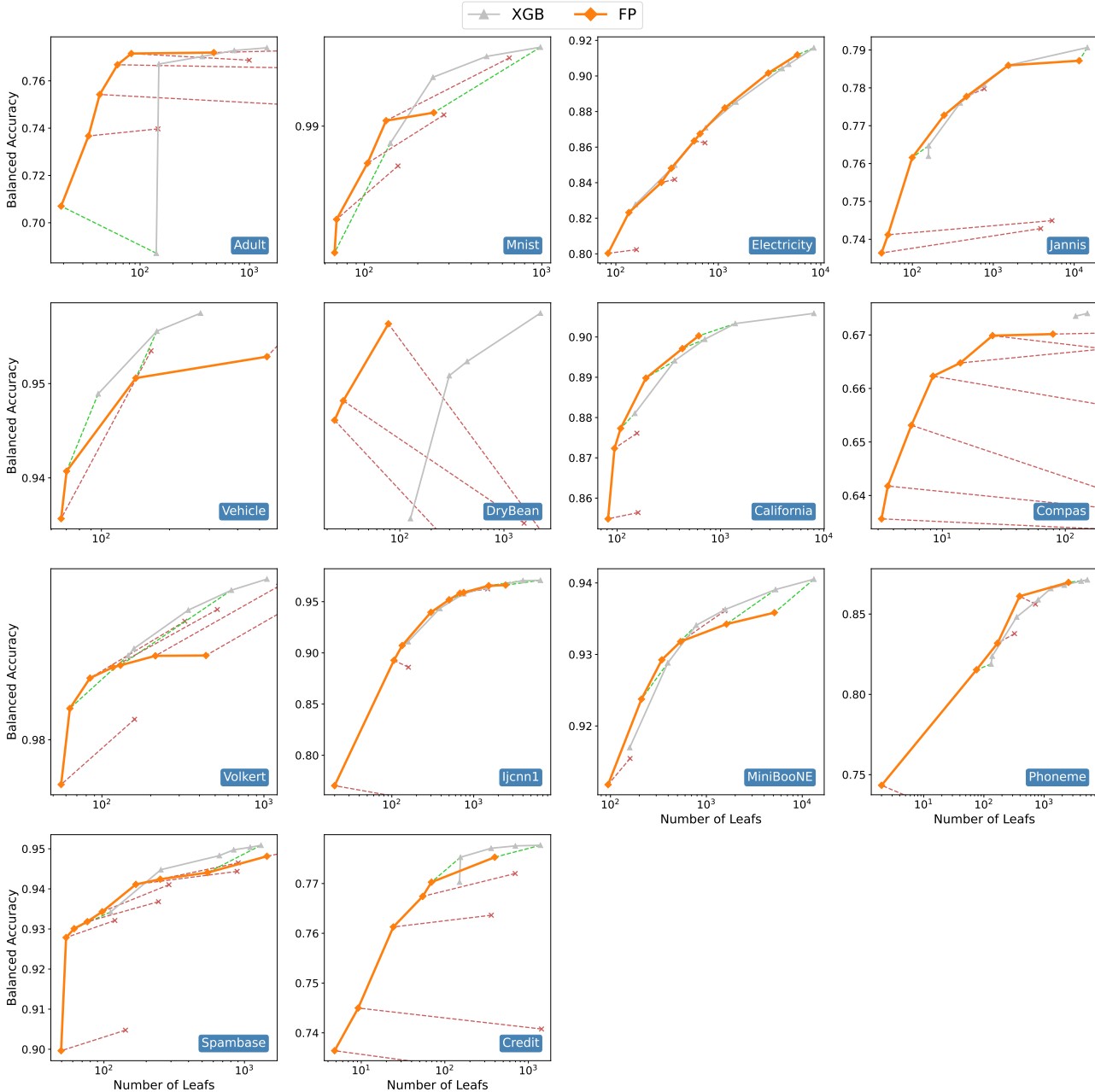

*Figure A6.* Comparison of Pareto fronts of FP vs. XGBoost on all datasets. The x-axis shows the number of leaves in log scale and the y-axis shows the balanced accuracy. The dashed lines connect a compressed FP model to the performance of an original XGBoost model. These are colored red if the original model is not on the Pareto front for XGBoost.

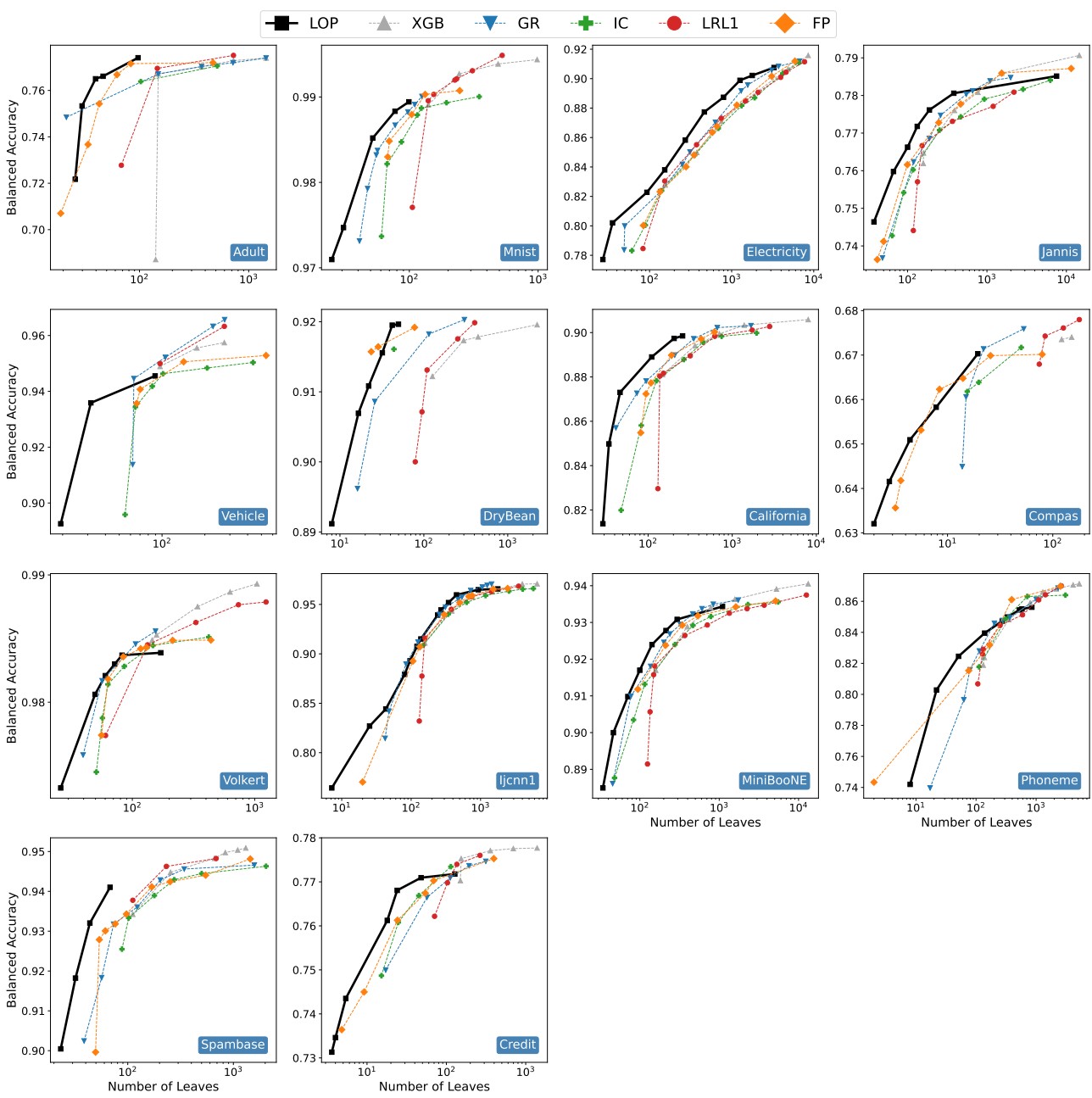

*Figure A7.* Pareto fronts for all compression methods on all datasets. The x-axis shows the number of leaves in log scale and the y-axis shows the balanced accuracy.

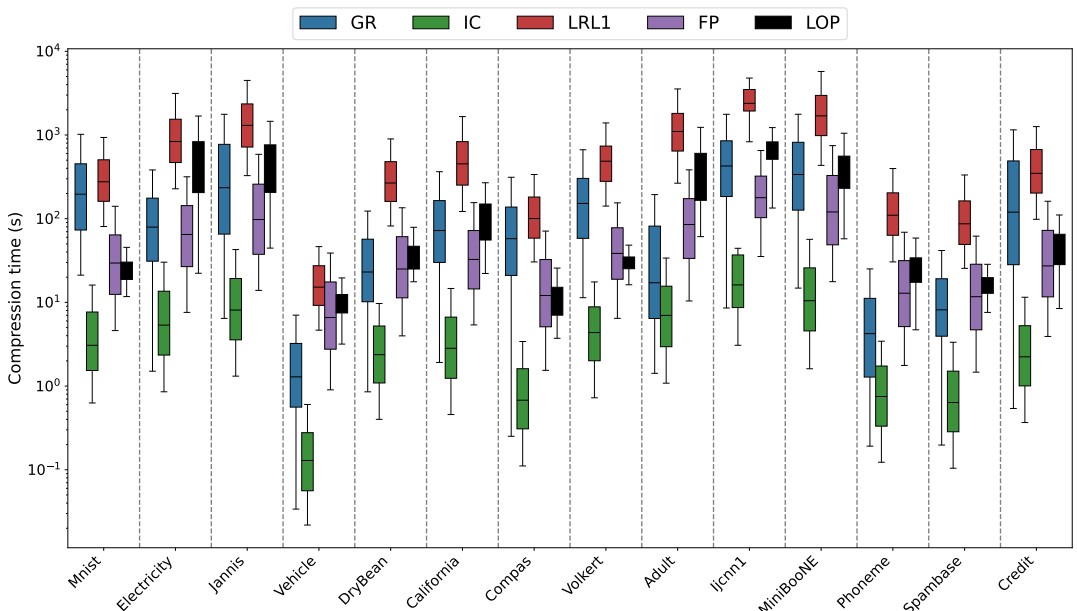

*Figure A8.* The distribution of the compression time (s) of XGBoost models in seconds on a log scale.

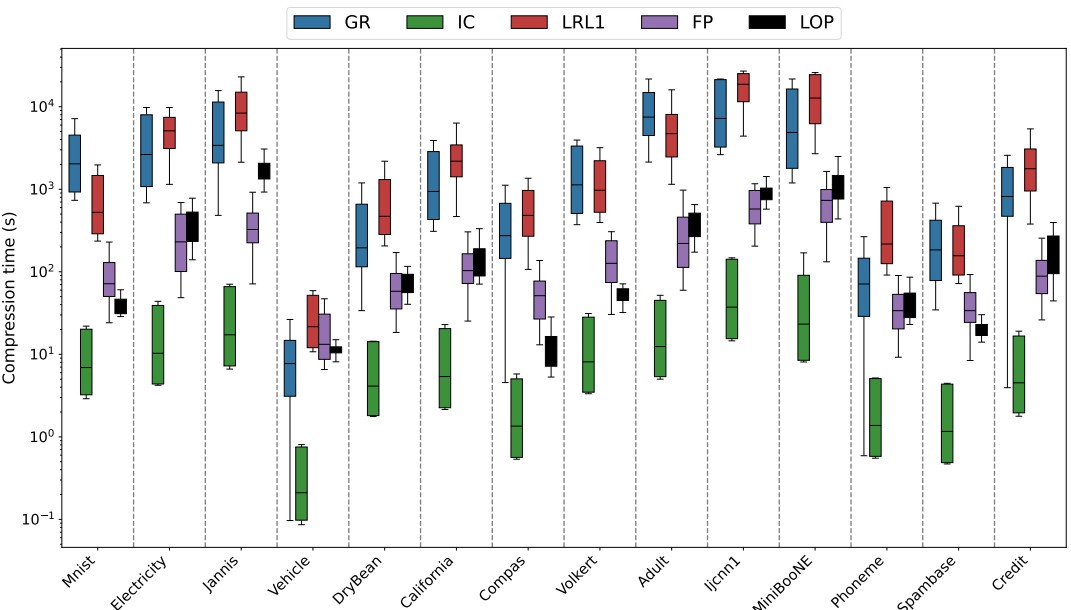

*Figure A9.* The distribution of the compression time (s) of RandomForest models in seconds on a log scale.

*Table A5.* The average tree reduction ratio for each dataset for the XGBoost models on the regression experiments. This ratio is computed as the number of trees in the original ensemble divided by the number of trees in the compressed model. Higher ratios are better. The best ratios are indicated in bold.

|  | GR | LRL1 | FP | LOP |
|---|---|---|---|---|
| Abalone | 1.0 | 1.5 | 11.2 | **25.9** |
| Ailerons | 1.0 | 1.4 | 4.6 | **12.7** |
| CpuSmall | 1.0 | 1.5 | 1.0 | **9.5** |
| Houses | 1.0 | 1.3 | 3.4 | **6.5** |
| House16H | 1.0 | 1.6 | 12.5 | **16.2** |
| WineQuality | 1.0 | 1.4 | 11.7 | **15.3** |
| Elevators | 1.0 | 1.2 | 1.3 | **6.2** |
| average | 1.0 | 1.4 | 6.5 | **13.2** |

*Table A6.* The average tree reduction ratio for each dataset for the RandomForest models on the regression experiments. This ratio is computed as the number of trees in the original ensemble divided by the number of trees in the compressed model. Higher ratios are better. The best ratios are indicated in bold.

|  | GR | LRL1 | FP | LOP |
|---|---|---|---|---|
| Abalone | 1.0 | 1.0 | 2.9 | **6.4** |
| Ailerons | 1.0 | 1.0 | **6.2** | 5.5 |
| CpuSmall | 1.0 | 1.0 | **8.4** | 6.6 |
| Houses | 1.0 | 4.2 | **12.0** | 9.9 |
| House16H | 1.0 | 5.6 | 3.7 | **9.0** |
| WineQuality | 1.0 | 2.4 | **5.6** | 4.8 |
| Elevators | 1.0 | 2.9 | **7.5** | 7.3 |
| average | 1.0 | 2.6 | 6.6 | **7.1** |

on all combinations of the following hyperparameters:

|  | XGBoost | RandomForest |
|---|---|---|
| $M$ | $\in [10, 25, 50, 100]$ | $\in [50, 100, 250]$ |
| $D$ | $\in [4, 6, 8]$ | $\in [10, 15]$ |
| $\eta$ | $\in [0.1, 0.25, 0.5, 1.0]$ | not applicable |

with $M$ the number of trees, $D$ the maximum depth of the trees and $\eta$ the learning rate in XGBoost. This yields 48 XGBoost models and 6 RandomForest models, to which we then apply the different compression algorithms. There are two differences to our binary classification setup in that we (1) evaluate the Root Mean Squared Error (RMSE) instead of balanced accuracy and (2) use a maximum allowed drop in performance $\Delta = 2\%$ that is relative to the uncompressed model's performance (e.g., if the uncompressed model achieves a RMSE of 0.250 on the validation set, then the compressed model's RMSE on that same validation set should be smaller than 0.255).

The left hand side of Tables A9 and A10 show the compression factors obtained by each compression algorithm by

averaging over all learned XGBoost and RandomForest per dataset. Similarly to the binary classification setting, LOP achieves the best compression factors, with FP and GR being second and third best. LOP's average compression factor ranges from 19 to 234 on XGBoost models and from 5 to 744 on RandomForest models.

The right hand side of Tables 1 and 2 show the RMSE for respectively the XGBoost and RandomForest models before and after compression. All compression methods yield an average performance loss on an independent test set that typically remains below the threshold $\Delta$. This indicates, as also found in the binary classification setting, that overfitting to the validation set is unlikely. Consequently, LOP achieves higher compression ratios for the same loss in predictive performance compared to the baselines.

We now discuss how the compression methods affect the number of trees retained in the ensembles. Tables A5 and A6 show the average tree reduction ratios for the different methods on XGBoost and RandomForest models, respectively. Higher numbers indicate fewer trees in the compressed model. On both XGBoost and RandomForest models, ensembles compressed by LOP have the fewest number of trees. This finding deviates slightly from the results on binary classification, where FP is more effective at pruning full trees in RandomForest models; see Table A4). The fact that FP prunes fewer trees than in the binary classification setting could explain why it achieves lower compression ratios for RandomForest regression.

**Q5: Scalability analysis.**

We now explore how compression approaches scale based on two ways of characterizing the size of the original ensemble: (1) $M$, the number of trees in an ensemble and (2) $D$, the maximum allowed depth of an individual tree. We consider the same four binary classification datasets that we used for Q3.

First, we explore varying $M \in [100, 250, 500]$ while keeping the maximum tree depth fixed. For XGBoost, we fix $D = 6$ and the learning rate $\eta = 0.1$ and for RandomForest we fix $D = 10$. Figures A12 and A13 show how the compression time in seconds varies as a function of $M$ for XGBoost and RandomForest, respectively. Note that time is on a log scale. Interestingly, LOP maintains a nearly constant compression time as the number of trees increases. This is likely because it is able to remove many trees upfront and this initial optimization problem is very simple. In contrast, the compression time for the other approaches increases as the number of trees in the original ensemble increases. Finally, LOP is typically the second faster approach after IC, however, IC achieves very little compression compared to LOP. In conclusion, LOP offers excellent scalability with respect to the number of trees in an ensemble compared to

*Table A7.* The average exact empirical robustness before (XGB) and after compressing the ensemble according to Kantchelian et al's (Kantchelian et al., 2016) exact MILP approach to find the nearest adversarial example for 500 randomly selected test examples. Results are averaged over the 48 binary classification XGBoost models. The best results are in bold.

|  | XGB | GR | IC | LRL1 | FP | LOP |
|---|---|---|---|---|---|---|
| Mnist | 0.194 | 0.195 | 0.214 | 0.180 | **0.221** | **0.221** |
| Electricity | 0.043 | 0.031 | 0.041 | 0.039 | 0.035 | **0.048** |
| Jannis | 0.067 | 0.097 | 0.098 | 0.079 | 0.096 | **0.108** |
| Vehicle | 0.145 | 0.147 | 0.158 | 0.138 | 0.155 | **0.162** |
| DryBean | 0.181 | 0.182 | 0.209 | 0.176 | **0.219** | 0.215 |
| California | 0.079 | 0.078 | 0.083 | 0.077 | 0.080 | **0.092** |
| Compas | 0.050 | 0.165 | 0.076 | 0.055 | **0.175** | 0.170 |
| Volkert | 0.164 | 0.187 | 0.196 | 0.178 | **0.198** | 0.197 |
| Adult | 0.348 | 0.338 | 0.410 | 0.341 | 0.449 | **0.469** |
| Ijcnn1 | 0.224 | 0.197 | **0.230** | 0.216 | 0.224 | 0.226 |
| MiniBooNE | 0.114 | 0.122 | 0.128 | 0.123 | 0.121 | **0.134** |
| Phoneme | 0.098 | 0.088 | 0.109 | 0.081 | **0.106** | 0.103 |
| Spambase | 0.091 | 0.091 | 0.100 | 0.087 | 0.114 | **0.120** |
| Credit | 0.112 | 0.275 | 0.207 | 0.132 | 0.283 | **0.301** |
| Average | 0.136 | 0.157 | 0.161 | 0.136 | 0.177 | **0.183** |

*Table A8.* Summary of which methods exceed the 6 hour timeout for Q1 for RandomForest (i.e., on how many folds out of 5). $M$ and $D$ denote the number of trees and depth of the trees in a forest.

|  | $M$ | $D$ | GR | LRL1 |
|---|---|---|---|---|
| Adult | 250 | 15 | 2/5 | NA |
| Ijcnn1 | 100 | 15 | NA | 5/5 |
| Ijcnn1 | 250 | 10 | 2/5 | 5/5 |
| Ijcnn1 | 250 | 15 | 4/5 | 5/5 |
| Jannis | 250 | 15 | NA | 5/5 |
| MiniBooNE | 250 | 10 | NA | 4/5 |
| MiniBooNE | 250 | 15 | 5/5 | 5/5 |

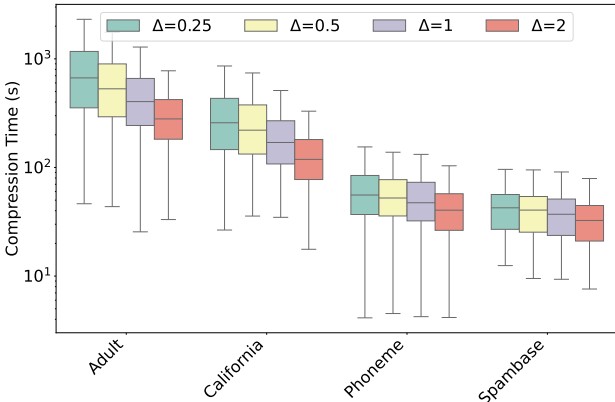

*Figure A10.* The distribution of compression time (in seconds) when varying LOP's hyperparameter $\Delta \in [0.25, 0.5, 1, 2]$

its competitors.

Second, we explore varying the maximum tree depth while keeping the number of trees in the ensemble fixed. For XG-Boost, we consider $D \in [6, 8, 10]$ with $M = 50$ and $\eta = 0.1$. For RandomForest, we consider $D \in [10, 15, 20, 25]$ with $M = 100$. Figures A14 and A15 show how the compression time in seconds varies as a function of $D$ for XG-Boost and RandomForest, respectively. Note that time is on a log scale and that some methods exceed the time limit; see Table A11 for an overview. For XGBoost, the maximum tree depth does not exhibit a strong influence on the compression time of any of the approaches. An interesting exception occurs with GR on XGBoost models for Adult: its runtime decreases substantially as the tree depth increases. This is likely because the compression operations that GR can perform yield larger than permitted performance drops meaning; hence it has a small runtime because it performs

little, if any, compression. For RandomForests, the story is slightly different. The runtime for LOP and FP tends to increase as the maximum tree depth does whereas this parameter has less effect on the other approaches. Still, this hyperparameter does not affect the relative order of runtime performance for the compression methods.

*Table A9.* **Compression results on XGBoost regression models.** The original models are characterized by their number of leaves (#Leaf) and root mean squared error (RMSE) on the test set. We report the compression ratios and test set RMSE for each compression technique. A $8\times$ compression ratio means that the compressed model has 8 times fewer leaf nodes than the original ensemble. The best compression ratios are given in **bold**.

| Dataset | #Leaf | Compress. ratio ($\times$) | | | | RMSE | | | | |
|---|---|---|---|---|---|---|---|---|---|---|
| | XGB | GR | LRL1 | FP | LOP | XGB | GR | LRL1 | FP | LOP |
| Abalone | 2557.6 | 28.6 | 1.4 | **178.3** | 120.4 | 0.298 | 0.297 | 0.308 | 0.303 | 0.293 |
| Ailerons | 3406.3 | 11.1 | 1.3 | 12.9 | **35.2** | 0.202 | 0.200 | 0.196 | 0.202 | 0.205 |
| Cpu | 2803.3 | 4.8 | 1.5 | 1.8 | **29.2** | 0.141 | 0.132 | 0.129 | 0.140 | 0.140 |
| Houses | 3810.4 | 8.4 | 1.3 | 6.9 | **19.6** | 0.173 | 0.173 | 0.167 | 0.175 | 0.176 |
| House16H | 3783.3 | 16.6 | 1.6 | 82.4 | **233.8** | 0.330 | 0.335 | 0.327 | 0.339 | 0.343 |
| WineQuality | 2969.0 | 16.8 | 1.3 | 71.6 | **234.6** | 0.352 | 0.353 | 0.355 | 0.351 | 0.358 |
| Elevators | 3795.2 | 8.4 | 1.2 | 2.2 | **21.6** | 0.213 | 0.207 | 0.202 | 0.213 | 0.219 |
| Average | 3303.6 | 13.5 | 1.4 | 50.9 | **99.2** | 0.244 | 0.243 | 0.240 | 0.246 | 0.248 |

*Table A10.* **Compression results on RandomForest regression models.** The original models are characterized by their number of leaves (#Leaf) and root mean squared error (RMSE) on the test set. We report the compression ratios and test set RMSE for each compression technique. A $8\times$ compression ratio means that the compressed model has 8 times fewer leaf nodes than the original ensemble. The best compression ratios are given in **bold**.

| Dataset | #Leaf | Compress. ratio ($\times$) | | | | RMSE | | | | |
|---|---|---|---|---|---|---|---|---|---|---|
| | RF | GR | LRL1 | FP | LOP | RF | GR | LRL1 | FP | LOP |
| Abalone | 79168.6 | 19.8 | 1.0 | 35.3 | **744.6** | 0.272 | 0.273 | 0.272 | 0.274 | 0.280 |
| Ailerons | 184674.7 | 2.2 | 1.0 | 6.2 | **7.7** | 0.185 | 0.185 | 0.185 | 0.189 | 0.190 |
| Cpu | 118589.5 | 1.6 | 1.0 | 8.3 | **10.1** | 0.118 | 0.118 | 0.118 | 0.122 | 0.125 |
| Houses | 300815.2 | **81.9** | 4.2 | 12.0 | 51.5 | 0.159 | 0.162 | 0.158 | 0.163 | 0.164 |
| House16H | 200964.8 | 105.6 | 5.3 | 3.7 | **492.5** | 0.306 | 0.316 | 0.313 | 0.311 | 0.327 |
| WineQuality | 62076.9 | **6.0** | 2.4 | 5.3 | 5.8 | 0.322 | 0.326 | 0.325 | 0.328 | 0.331 |
| Elevators | 189162.0 | **93.9** | 2.9 | 7.4 | 23.6 | 0.204 | 0.203 | 0.206 | 0.206 | 0.207 |
| Average | 162207.4 | 44.4 | 2.5 | 11.2 | **190.8** | 0.224 | 0.226 | 0.225 | 0.227 | 0.232 |

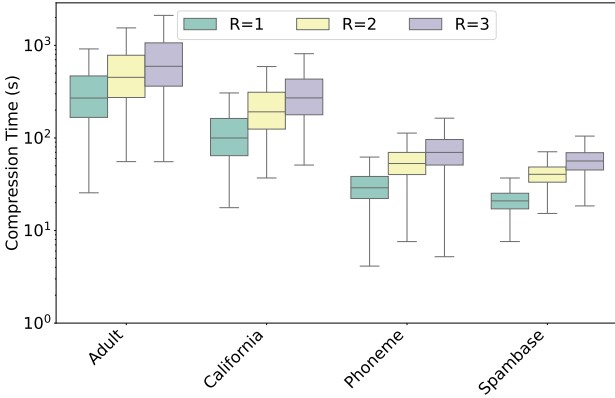

*Figure A11.* The distribution of compression time (in seconds) when varying LOP's hyperparameter $R \in [1, 2, 3]$.

*Table A11.* Summary of which methods exceed the 6 hour timeout for Q5 (scalability) for RandomForest (i.e., on how many folds out of 5). $M$ and $D$ denote the number of trees and depth of the trees in a forest.

| | $M$ | $D$ | GR | LRL1 |
|---|---|---|---|---|
| Adult | 100 | 15 | 3/5 | NA |
| Adult | 100 | 20 | 2/5 | NA |
| Adult | 100 | 25 | 3/5 | 4/5 |
| Adult | 250 | 10 | 5/5 | 1/5 |
| Adult | 500 | 10 | 5/5 | 5/5 |

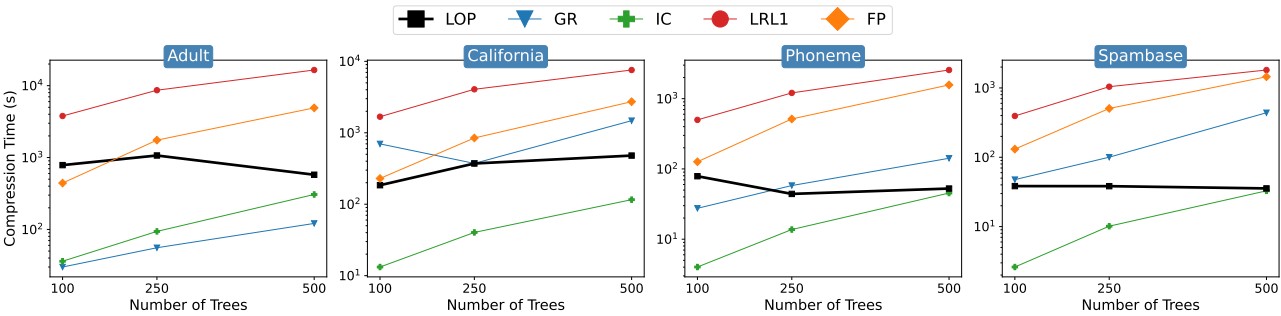

*Figure A12.* Average compression time (s) as a function of the number of trees $M$ in the original XGBoost models with $D = 6$ and $\eta = 0.1$. Results are shown for each compression method on four representative datasets.

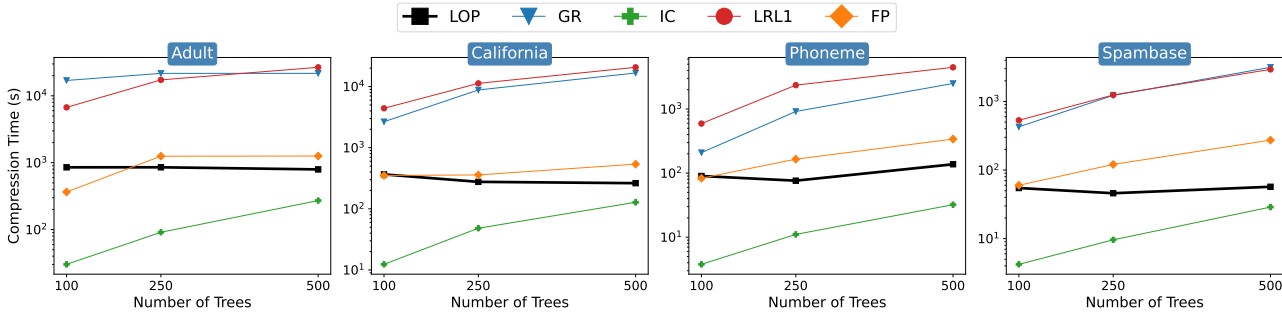

*Figure A13.* Average compression time (s) as a function of the number of trees $M$ in the original RandomForest models with $D = 10$. Results are shown for each compression method on four representative datasets.

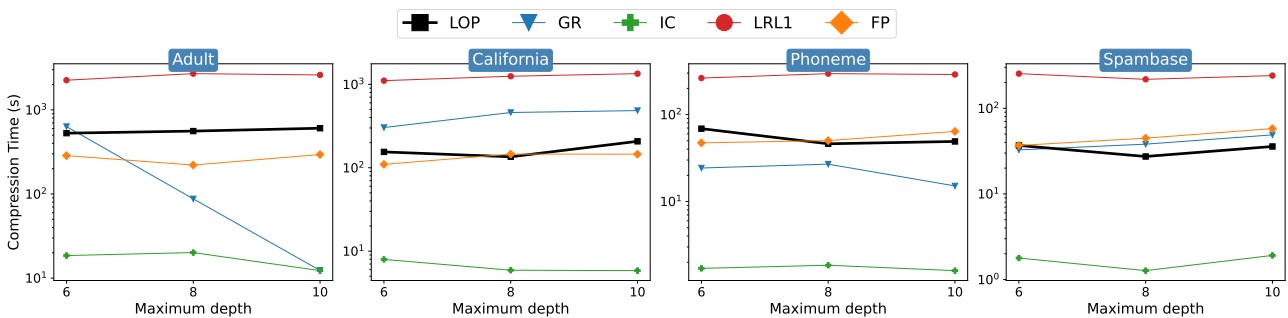

*Figure A14.* Average compression time (s) as a function of the maximum tree depth $D$ used when learning the original XGBoost models with $M = 50$ and $\eta = 0.1$. Results are shown for each compression method on four representative datasets.

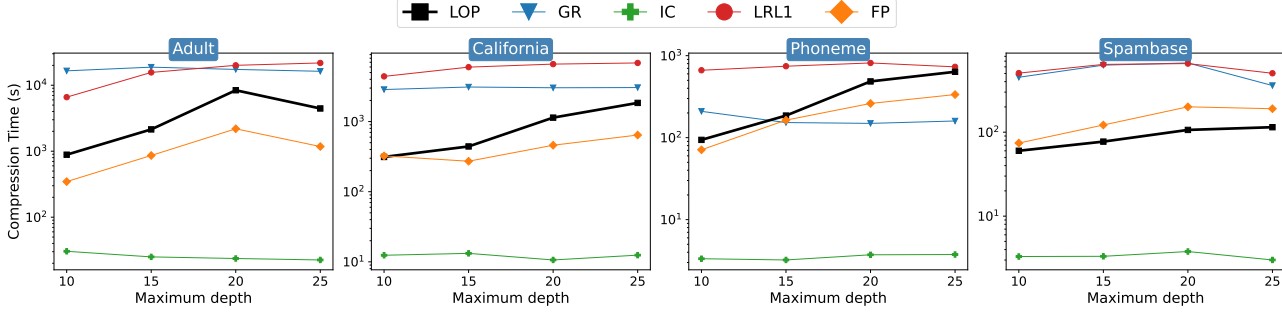

*Figure A15.* Average compression time (s) as a function of the maximum tree depth $D$ used when learning the original RandomForest models with $M = 10$. Results are shown for each compression method on four representative datasets.

