# OpenReview forum: "Compressing tree ensembles through Level-wise Optimization and Pruning"
_ICML.cc/2025/Conference — ICML 2025 poster_

### Official Review · Reviewer_CuzX · 2025-03-08

**Overall Recommendation:** 3

**Summary:**

This paper proposes a new algorithm for compressing a learned tree ensemble while keeping its generalization performance.
In each depth of a given tree ensemble, the proposed method prunes its redundant subtrees and adjusts the remaining leaf values.
Through the experiments on binary classification datasets, the authors demonstrated that the proposed method attained higher compression rates than the existing baseline methods without significantly degrading accuracy.
In addition, the experimental results showed that the proposed method could improve the computational costs of both test inference and robustness verification.

## update after rebuttal
I appreciate the authors for their insightful response. Because the additional experimental results provided by the authors address my concern regarding the lack of comparisons with the existing methods, I have decided to improve my score. If the paper is accepted, I hope the authors will include these results in the final version.

**Claims And Evidence:**

My main concern is the lack of comparisons to the existing methods, such as [Nan+, NeurIPS2016] and [Liu+, AISTATS2023], that aim to reduce the complexity of tree ensembles by pruning each tree.
I believe the novelty and effectiveness of the proposed method can not be supported without comparisons with these existing baselines.

- [Nan+, NeurIPS2016] Feng Nan, Joseph Wang, Venkatesh Saligrama. Pruning Random Forests for Prediction on a Budget. NeurIPS, 2016.
- [Liu+, AISTATS2023] Brian Liu, Rahul Mazumder. ForestPrune: Compact Depth-Pruned Tree Ensembles. AISTATS, 2023.

**Essential References Not Discussed:**

As mentioned above, the key contributions of this paper seem to be related to the work by [Nan+, NeurIPS2016] and [Liu+, AISTATS2023].
I also think this paper is related to the existing studies on extracting rules from tree ensembles from the perspective of interpretability, e.g., [Hara+, AISTATS2018] and [Liu+, KDD2024].

**Experimental Designs Or Analyses:**

- My main concern is the lack of comparisons to the related baselines ([Nan+, NeurIPS2016] and [Liu+, AISTATS2023]), as mentioned above.
- Another concern is that the number of trees and maximum tree depth were only examined up to 100 and 8, respectively. Since the computational complexity of the proposed method depends on these parameters, I think the scalability and sensitivity of the proposed method with respect to these parameters should be investigated.

**Methods And Evaluation Criteria:**

- I think the proposed representation $c_n v_k + b_n$ is an interesting idea that can simultaneously express pruning a redundant subtree or refinements of leaf values.
- I also think the experimental results shown in Table 2 demonstrated well the effectiveness of the proposed method in terms of not only compression performance but also verification efficiency. However, as mentioned above, I am concerned about the lack of comparisons to the related baselines.

**Other Comments Or Suggestions:**

Nothing in particular.

**Other Strengths And Weaknesses:**

I believe the presentation of this paper could be improved.
For example, this paper seems to use parentheses too frequently, which makes some sentences complex and difficult to follow.

**Questions For Authors:**

Nothing in particular.

**Relation To Broader Scientific Literature:**

The key contributions of the paper are related to the field of practical techniques for learning tree ensembles. In particular, pruning a learned tree ensemble is one of the promising approaches from the perspectives of generalization performance [Ren+, CVPR2015], computational efficiency [Nan+, NeurIPS2016] [Liu+, AISTATS2023], and interpretability [Hara+, AISTATS2018] [Liu+, KDD2024].

- [Hara+, AISTATS2018] Satoshi Hara, Kohei Hayashi. Making Tree Ensembles Interpretable: A Bayesian Model Selection Approach. AISTATS, 2018.
- [Liu+, KDD2024] Brian Liu, Rahul Mazumder. Fire: An Optimization Approach for Fast Interpretable Rule Extraction. KDD, 2024.

**Theoretical Claims:**

This paper does not include theoretical claims.

---

> ### Author Rebuttal · Authors · 2025-04-01
>
> Many thanks for your insightful comments. Below, we address your main concerns: (a) baselines for comparison and (b) scaling behavior.
>
> Existing baselines:
> Thank you for pointing us to this related work. Among the listed papers, we find Nan et al. (2016) less relevant as it solves a different problem: obtaining the value of each feature has a cost and the goal is to minimize the expected feature acquisition cost at prediction time.  This is a very different goal. The other papers are more relevant and we will discuss them in the revised version (it seems we missed them because they are not connected through citations with the body of literature we studied or the baselines we compare to, so thank you for pointing us to them).
>
> The novelty of our work is not jeopardized, as all these methods clearly differ from our approach:
>
> - ForestPrune (Liu et al. 2023) simplifies forests by cutting whole trees at a specific level, rather than pruning individual nodes like LOP does. This way, it loses a crucial aspect of trees, namely, that some subtrees can be deeper than others: a tree can partition the input space in a finer-grained manner in some areas, and in a coarser manner elsewhere. A second difference is that LOP refines leaf values, while ForestPrune does not. Hence, LOP explores ensembles that are not in ForestPrune’s search space.
>
> Empirically, we have compared LOP and ForestPrune on XGBoost classifiers on all datasets mentioned in our paper. We have taken the same XGB settings, that is, number of trees in [10, 25, 50, 100], tree depth in [4, 6, 8] and learning rate in [0.1, 0.25, 0.5, 1] and averaged the results. As the table below shows, LOP’s compression factors are up to 10 times better than ForestPrune’s, for comparable accuracy:
>
> | | Compr.: | LOP   | forestprune  | Diff. BAcc.: | LOP  | forestprune |
> |-----------------|--------|------:|-------------:|-----|-----:|-----------: |
> |Spambase || 8.5   | 3.6  || 0.9  | 0.7 |
> |Phoneme || 7.5   | 3.7  || 1.3  | 0.9 |
> |Electricity || 5.2   | 1.4  || 0.5  | 0.2 |
> |Adult || 31.8  | 37.2 || -0.2 | 0.1 |
> |Credit || 196.6 | 133.2|| 0.5  | 0.4 |
> |CompasTwoYears  || 356.8 | 240.3|| -0.1 | -0.5 |
> |DryBean[6vRest] || 28.8  | 15.6 || 0.3  | 0.4 |
> |Mnist[2v4] || 8.0   | 3.1  || 0.6  | 0.4 |
> |Volkert[2v7] || 8.3   | 4.9  || 0.6  | 0.4 |
> |Jannis  || 18.1  | 15.3 || 0.6  | 0.5 |
> |Vehicle || 8.0   | 2.6  || 1.4  | 1.8 |
> |MiniBooNE || 6.2   | 2.6  || 0.6  | 0.4 |
> |California || 9.2   | 2.5  || 0.6  | 0.4 |
> |Ijcnn1 || 6.5   | 2.3  || 0.4  | 0.1 |
> |Average || 50.0  | 33.4 || 0.6  | 0.4 |
>
> In terms of runtime, compressing an ensemble takes LOP on average 236.5 seconds, whereas ForestPrune takes 98.5 seconds. We will include the full results in a revised version of the paper.
>
> - Hara et al. (2018) learn an additive model that contains a small set of “rules”; a rule is one path from root to leaf. This corresponds to an ensemble where each tree is constrained to have 1 leaf with non-zero value. LOP has no such constraint. Their experiments only consider models with at most 10 rules.
>
> - FIRE (Liu et al. 2024) re-learns leaf weights, regularizing for sparsity and “fusion” of non-zero leaves. It is very similar to the Global Refinement method (Ren et al. 2015), which we do cite and compare to in our paper; the main difference between GR and FIRE is in the fusion regularization and in the optimizer. LOP uses a different parametrization than FIRE: it uses fewer variables, yielding simpler optimization problems to be solved.
>
> Scaling:
> We only considered up to 100 trees because it is known that including more trees rarely improves predictive performance. Similarly, XGBoost may overfit with deeper trees. We felt that testing the method on larger ensembles would artificially boost compression factors. That being said, we ran two experiments.
>
> 1. We trained XGBoost with 100, 250 and 500 trees and max depth 8 on four datasets: Adult, California, Spambase, and Phoneme. LOP scales well. For LOP, the average run time on 100 trees is 195s, 250 trees is 520s and 500 trees is 425s. We hypothesize that adding more trees allows LOP to prune full trees more aggressively, leading to more efficient pruning on lower levels. For GR, average run time is 225s for 100 trees, 436s for 250 trees and 1988s for 500 trees. For LRL1 it is 2704s, 3667s, 7581s.  For IC it is 9s, 45s and 128s. All times are in seconds.
> The average compression ratios are 86.4 for LOP, 4.6 for GR, 3.8 for LRL1 and 4.9 for IC.
>
> 2. For scaling depth, the response to reviewers N4NE/fAT9 shows results for Random Forests with up to 500 trees and depth 15. ForestPrune fails to compress the biggest models, in which case it returns the empty model. We hypothesise that this is due to the greedy bottom-up approach of ForestPrune: it starts with the empty model and adds (partial) trees incrementally. We will investigate this further.
>
> We will add these results to the paper.

---

> > ### Comment · Reviewer_CuzX · 2025-04-02
> >
> > Thank you for your insightful rebuttal. The additional experimental results were quite interesting. I have decided to improve my score.

---

> > > ### Author Response · Authors · 2025-04-04
> > >
> > > We really appreciate that you have taken the time to read our rebuttal. In particular, we are happy that you found our additional results interesting and that it has improved your opinion of the work. We are wondering if anything else could be further addressed to make our paper even more convincing.

---

### Official Review · Reviewer_sKmw · 2025-03-10

**Overall Recommendation:** 4

**Summary:**

The manuscript proposes a novel method (LOP) for compressing tree-based ensembles by pruning leaves and/or entire trees. LOP is based on sparse optimization and has applications to bagging or boosting ensembles. Experimental results show that the approach cuts down model size with minimal impact on performance. This makes models more energy-efficient and easier to formally verify.

## update after rebuttal

The new experiments in reply to other reviewers' comments only reaffirm my view that LOP is a fast and effective method for compressing tree-based models. I think this manuscript makes a meaningful contribution to the literature on this topic, and therefore stand by my original assessment.

**Claims And Evidence:**

The primary claim of the manuscript is that LOP reduces model size without harming performance (too much). This is verified in a wide range of experiments on benchmark datasets. Three relevant baselines are considered (GR, IC, LRL1). LOP fares well in all trials.

**Essential References Not Discussed:**

I'm unaware of any essential references that were omitted.

**Experimental Designs Or Analyses:**

The experimental design is sensible and rigorous. I especially like the Pareto frontier plots.

**Methods And Evaluation Criteria:**

The selection of datasets and benchmarks are sensible. The experiments appear well-designed. The results are clear and compelling.

**Other Comments Or Suggestions:**

-p. 7: "LRL1 is by clearly the slowest approach" -> "LRL1 is clearly" or "LRL1 is by far..."

-In Tables 1 and 3, it would be helpful to bold the winning results as elsewhere.

**Other Strengths And Weaknesses:**

The writing is clear and direct. The proposal is not exactly revolutionary, but the experimental results clearly suggest that LOP is effective at reducing model size with minimal information loss. This is an important contribution.

**Questions For Authors:**

N/A

**Relation To Broader Scientific Literature:**

The work connects with recent efforts to distill ensemble models into more compressed representations.

**Theoretical Claims:**

The manuscript does not make any theoretical claims.

---

> ### Author Rebuttal · Authors · 2025-04-01
>
> Many thanks for your positive comments! We will take your suggestions into account.

---

### Official Review · Reviewer_N4NE · 2025-03-12

**Overall Recommendation:** 4

**Summary:**

The paper "Compressing tree ensembles through Level-wise Optimization and Pruning" proposes a combination of ensemble pruning, decision tree pruning and leaf-refinement to reduce the memory footprint of forests for reduced resource usage during deployment. To do so, the authors re-formulate the inference of a forest into a tensor-based formulation that also captures the inferencing process on each level of each tree in the forest. Based on this formulation, a pruning algorithm is presented, that starts at the root node of each tree and iteratively refines labels while removing subtrees. For level $l=0$ this means removing entire trees, while at the last level we remove leaf nodes. In-between, partial trees are removed. The experimental evaluation on 14 datasets shows that the pruned forests are comparable (in a 0.5% environment) in performance, while being much, much smaller.

**Claims And Evidence:**

The paper is generally well-written and the arguments of the authors can be easily followed. While I do have some minor questions wrt. to the experimental evaluation, I think the authors present convincing evidence for each claim.

**Essential References Not Discussed:**

I am not aware of any missing essential references.

**Experimental Designs Or Analyses:**

The experimental design is overall sound. Direct competitors are compared over 14 benchmarking datasets. The runtime and memory footprint is included. While I generally prefer larger sets of experiments for tree ensembles (i.e. more datasets and comparisons via critical difference diagrams), I think the overall analysis still holds.

**Methods And Evaluation Criteria:**

Although somewhat limited (14 datasets, 5 methods), the experimental evaluation fits the proposed method.

**Other Comments Or Suggestions:**

In case the paper is not accepted, I suggest improving the following parts:
- Personally, the example in section 3.0 and 3.1 was not really helpful to me. I would have preferred more explanations on the method itself
- Neither $\alpha$, nor $\Delta$ are explained in detail. While the meaning of both parameters are somewhat clear (see my question below), they are detrimental for Alg 1. I would suggest adding more explanations about that
- I think Q3 in the experimental evaluation is slightly misleading: While it is true, that LOP has only two hyperparameters $\Delta$ and $R$, the overall approach has more hyperparameters such as learning rate, optimization method, loss etc. While I understand that these are "additional" hyperparameters, a practitioner has to choose these as well. I would add more discussions about this.

**Other Strengths And Weaknesses:**

The paper is generally well-written (minus a few questions I state below) and was easy to follow for me. I think, the formulation of the forest in tensor-notation is generally helpful given the current trend in tensor-based computations. The appendix A offers an example of when leaf-refinement is useful, which is nice for future reference.
The experimental evaluation could be enhanced by including more datasets and other methods, however, given that this is a conference submission I don't see any issue here (i.e. more is always better, but not always necessary).

**Questions For Authors:**

1) In the experimental evaluation, you state that you use a 5-fold cross-validation, with 3 folds for training, 1 for validation and 1 for testing. For fitting $\alpha$ you also mention that you are using a validation set. I am a bit confused by this. Typically, I would expect a three-way split: train data (for the forest), prune data (for running LOP) and then test data for testing. Now I would repeat this splitting X times with X different random seed. Please explain how you did this here?
2) Algorithm 1 shows that we have to perform an optimization problem $R\cdot d$ times (line 6). You mention that these are comparably simple / easy and the runtimes you show reflect this, but could you please comment on the exact sizes of these optimization problems. On $l=0$, I have to optimize over 2M parameters and then on the next level $2\cdot 2\cdot M$ (given all trees are balanced and have the same height) and so on?
3) How did you choose $\alpha$?
4) Do you have plans to make your implementation publicly available?
5) The "Joint leaf-refinement and ensemble pruning through l1 regularization" uses Random Forest while you use XGBoost. Why? And, did you try RF as well?

**Relation To Broader Scientific Literature:**

The paper presents a non-trivial, but somewhat natural extension of the recent "Joint leaf-refinement and ensemble pruning through l1 regularization" paper by Buschjäger and Morik from 2023. While the original paper combines leaf-refinement with ensemble pruning, the method presented in this paper goes one step further and allows for pruning at every stage/level of every tree in the forest. Personally, I see that there is a growing (re-)interest in tree/ensemble pruning in the literature, and hence this paper fits perfectly. Due to its strong performance, it could become one of the go-to papers in this area of research.

**Theoretical Claims:**

No theoretical claims are made.

---

> ### Author Rebuttal · Authors · 2025-04-01
>
> Thank you for your positive comments, and especially that this paper “could become the go-to paper in this area”, and “more [experiments] is always better, but not always necessary”. Thanks also for the suggestions for improvement, which we will take into account.
>
> To answer your questions:
>
> Q1, Q3: The 5 fold cross-validation and tuning of alpha work as follows. We partition into 5 subsets. In each fold, 1 subset serves as test set. Of the 4 remaining subsets, 3 are used for training and pruning the ensemble (that is, X in Eq. 2 is based on these 3 subsets). The fourth 'validation' set is used to tune alpha; a model with performance drop $>\Delta$ is discarded. A log-range of alphas (range 0.001 to 10000) is tried, and among the models not discarded, the smallest model is returned. This is the model for which we ultimately report size, accuracy on test set, etc.
>
> Q2: The number of parameters to optimize at level 0 is M (not 2M because the b’s are excluded on this level, see lines 142-143, right column). At level 1, the number of parameters is not 2\*2\*M but 2\*2\*M’ with M’ the number of trees not pruned at level 0 (M’<=M). On each level we try to prune, and each pruned subtree becomes a leaf that contributes only 1 parameter to all lower levels, rather than 2, 4, 8… as we go deeper. Generally, at level l, if there are x active nodes (of which y leaves-on-a-higher-level and z internal nodes), there are y+2\*z < 2x parameters. By the time we proceed to the lowest level d, this number can be 2\*2^d\*M in the worst case, if nothing ever gets pruned, but the point is of course that we do prune a lot, starting at the upper levels.
>
> Q3: See Q1.
>
> Q4: Yes, the implementation is publicly available.
>
> Q5: We used XGBoost because we consider it the state of the art. In response to the review, we have run additional experiments with Random Forests. We also include results for ForestPrune (Liu et al. 2023) suggested by reviewer CuzX. Specifically, we use sklearn RFs with a max depth in [10, 15] and number of trees in [100, 250, 500], on 3 datasets. The table below shows those results. LOP averages a compression ratio of 228.2, whereas GR achieves a ratio of 120.2 and both IC and LRL1 achieve a ratio of 1.3. The average drop in balanced accuracy for LOP is 0.01, similar to the baselines. ForestPrune fails to compress the biggest models; it returned the empty model. Over the successful runs, ForestPrune has a compression ratio of 5 and loses 0.09 in balanced accuracy. In terms of run time, it takes LOP on average 164s to compress these ensembles, compared to 6892s for GR, 5072s for LRL1, and 2665s for ForestPrune. While IC is faster (72s), it has much worse compression ratios than LOP. Hence, LOP compresses more than the competitors without losing accuracy. Moreover, it is around 42x faster than GR which is its nearest competitor in terms of compression ratio. We will add these results to the paper.
>
> | Dataset | depth | nb_trees | Compr.: GR | IC | LRL1 | LOP | ForestPrune |Runtime: GR | IC| LRL1| LOP | ForestPrune |
> |------------|-------|----------|--------------:|------:|--------:|-------------:|---------:|----------------:|-------:|-------------:|------:|-------:|
> | California |10|100|120.7|1.1|2.6|166.6|2.5|3752.7|14.9|3209.2|213.0|562.6|
> | California ||250|210.8|1.1|1.0|396.5|2.7|13884.5|82.7|7718.6|235.1|2844.8|
> | California ||500|254.8|1.1|1.0|788.7|2.5|33363.5|352.2|18589.8|305.2|6030.0|
> | California |15|100|178.2|1.1|1.0|89.0|6.8|3358.0|19.6|5350.0|370.4|788.7|
> | California ||250|370.9|1.2|1.0|418.8|6.7|13987.4|105.5|12709.3|404.8|4385.3|
> | California ||500|564.1|1.2|1.0|533.5|6.8|32928.4|447.2|24220.1|446.9|9698.1|
> | Spambase |10|100|32.5|1.1|1.2|71.8|3.8|392.7|3.4|316.2|39.8|161.1|
> | Spambase || 250 | 52.7 | 1.1 | 1.0| 214.0 | 3.5 | 1514.8 | 12.8 | 778.2 | 44.1 | 951.4 |
> | Spambase || 500 | 81.5 | 1.2 | 1.0 | 405.0 | (fail) | 4978.8 | 44.7 | 1932.3 | 44.5 | 3478.4 |
> | Spambase | 15| 100 | 64.8 | 1.2 | 1.0 | 128.6 | 5.8 | 597.5 | 3.5 | 357.2 | 58.9 | 246.9 |
> | Spambase || 250 | 71.6 | 1.1 | 1.0 | 148.2 | 5.5 | 1833.6 | 14.7 | 1233.3 | 59.4 | 1478.6 |
> | Spambase || 500 | 86.9 | 1.2 | 1.0 | 390.1 | (fail) | 5134.0 | 54.2 | 2590.2 | 58.0 | 5122.1 |
> | Phoneme | 10 | 100 | 9.2 | 1.6 | 2.2 | 21.0 | 5.3 | 208.0 | 3.4 | 466.2 | 70.0 | 174.5 |
> | Phoneme | | 250 | 13.9 | 1.6 | 1.7 | 54.6 | (fail) | 981.1 | 13.7 | 1734.5  | 72.7  | 1026.4 |
> | Phoneme | | 500 | 21.6 | 1.6 | 1.3 | 162.3 | (fail) | 3771.3 | 45.5 | 3193.5 | 93.1 | 3650.6 |
> | Phoneme | 15 | 100 | 6.2 | 1.6 | 1.9 | 17.1 | 8.2 | 173.8 | 3.9 | 569.0 | 131.0 | 273.3 |
> | Phoneme | | 250 | 7.4 | 1.6 | 1.9 | 25.3 | (fail) | 626.9 | 15.0 | 2130.4 | 154.1 | 1571.6 |
> | Phoneme | | 500 | 15.3  | 1.6 | 1.1 | 77.0 | (fail) | 2577.8 | 51.8 | 4194.9 | 150.2 | 5523.0 |
> | Average ||| 120.2 | 1.3 | 1.3 | 228.2 | 5.0  | 6892.5 | 71.6 | 5071.8 | 164.0 | 2664.9 |

---

> > ### Comment · Reviewer_N4NE · 2025-04-03
> >
> > Thank you for taking the time to answer all of our questions, and especially thank you for running additional experiments.
> >
> > Just to confirm, you prune on the training set, i.e. there is no dedicated pruning set?
> >
> > In any case, I still recommend accepting the paper. In case the paper is accepted, I recommend updating the camera-ready version to include the additional explanation on the 3-way split during cross-validation as well as some additional information on the number of parameters (maybe a paragraph in the appendix is enough). Finally, I would mention your results on using RF and also possibly place them in the appendix.

---

> > > ### Author Response · Authors · 2025-04-04
> > >
> > > Thank you for responding. We will absolutely update a potential camera-ready version to have all of this additional information & results.
> > >
> > > We do indeed prune on the training set. This can be justified by the fact that the optimisation problem is different during training (i.e., optimising for balanced accuracy per tree) and pruning (i.e., a trade-off between balanced accuracy of the ensemble as a whole and the ensemble size, which uses a regulariser that was ignored during training). Therefore, a separate pruning set is not required.

---

### Official Review · Reviewer_fAT9 · 2025-03-13

**Overall Recommendation:** 2

**Summary:**

The paper introduces LOP (Level-wise Optimization and Pruning), a method for compressing decision tree ensembles by pruning subtrees level by level while updating leaf values to maintain predictive accuracy. Unlike prior methods that prune entire trees or merge only leaf nodes, LOP can remove subtrees at any level, achieving compression rates 10 to 100 times higher than competitors. By optimizing leaf values globally, it minimizes accuracy loss, typically staying within 0.5% of the original model. Empirical results show that LOP significantly reduces memory footprint, speeds up robustness verification, and enhances efficiency on resource-constrained devices, making tree ensembles more practical for deployment.

## update after rebuttal
Thank you to the authors for providing detailed feedback and clarifications. While the responses are appreciated, they do not sufficiently address my primary concerns or significantly alter my overall evaluation of the manuscript. Therefore, I will maintain my original score, and I strongly recommend a major revision to strengthen the paper.

**Claims And Evidence:**

The paper’s central claims are well supported by empirical evaluations on multiple datasets and rigorous comparisons with baseline compression methods. Specifically, the assertion that LOP can achieve compression rates 10 to 100 times greater than existing techniques while maintaining nearly the same predictive performance is backed by quantitative results across 14 benchmark datasets, demonstrating high compression ratios with minimal accuracy loss. Additionally, the paper provides strong evidence that robustness verification is significantly faster on LOP-compressed models compared to both the original XGBoost models and those compressed by competing methods. The authors further substantiate LOP’s efficiency gains with explicit memory footprint calculations via proxies, highlighting its advantages in resource-constrained environments. However, one limitation is that the experiments are confined to binary classification tasks, leaving open the question of whether LOP’s compression benefits extend to multi-class problems or other data modalities such as image data.

**Essential References Not Discussed:**

While the paper covers traditional ensemble pruning techniques, it overlooks more recent advancements, such as knowledge distillation methods introduced by Hinton, Vinyals, and Dean (2015) in Distilling the Knowledge in a Neural Network. This technique, which transfers knowledge from a large model to a smaller one while preserving performance, has been adapted for tree ensembles, including gradient boosting distillation.

**Experimental Designs Or Analyses:**

The experimental design in the paper appears well-structured, with a strong benchmarking approach against relevant baselines. However, there are a few potential concerns:

- The evaluation is entirely on binary classification tasks, leaving out multi-class classification and regression, which could behave differently under compression. It is unclear whether LOP’s effectiveness generalizes to tasks with continuous target variables or high-cardinality categorical targets.
- The chosen baselines (GR, IC, and LRL1) are reasonable, but other pruning techniques, such as cost-complexity pruning in decision trees or neural network-inspired distillation methods, are not considered.
- The authors conduct a sensitivity analysis on $\Delta$ (allowed accuracy loss) and $R$ (number of pruning rounds), which is good practice. However, their choice of hyperparameters for the base XGBoost models (e.g., number of trees, learning rate, tree depth) is based on a predefined grid, which may not always yield the best initial models before compression. A stronger baseline might slightly alter results.
- Since LOP makes significant changes to the model structure, it would be useful to see results on unseen test distributions to test robustness.

**Methods And Evaluation Criteria:**

The proposed LOP method and the evaluation criteria are well-aligned with the problem of compressing tree ensembles while maintaining predictive performance. The authors assess LOP using 14 benchmark datasets from OpenML, covering diverse binary classification domains, which is appropriate given that tree ensembles like XGBoost and Random Forests are widely used for structured data tasks. The evaluation metrics—compression ratio, accuracy retention, robustness verification speed, and memory footprint—are relevant for assessing the trade-offs between model size and performance.

Moreover, comparisons against three competitive baselines (Global Refinement, Individual Contribution, and Leaf refinement combined with L1 ensemble pruning) ensure a fair assessment of LOP’s effectiveness. The use of Pareto front visualizations to illustrate the trade-off between compression and accuracy further strengthens the evaluation. However, a potential limitation is that all datasets are binary classification tasks, leaving unanswered how LOP would perform in multi-class classification or regression settings. Additionally, while LOP is tested on XGBoost ensembles, further validation on Random Forests or other tree-based architectures could enhance generalizability.

**Other Comments Or Suggestions:**

A deeper analysis of how accuracy degrades as more pruning is applied (beyond the Pareto front visualization) would provide useful insights for practitioners.

**Other Strengths And Weaknesses:**

While the paper discusses memory footprint and decision path length, it does not directly measure inference latency. Including actual prediction time benchmarks on different hardware (e.g., CPU, embedded systems) would provide stronger evidence of real-world efficiency gains.

**Questions For Authors:**

**Question 1:** Have you tested LOP on multi-class problems or regression tasks? If not, do you anticipate any fundamental challenges in applying LOP to these settings?

- If LOP does not generalize well beyond binary classification, its applicability would be more limited than suggested. Demonstrating strong performance in multi-class or regression tasks would increase confidence in LOP’s generalizability.

**Question 2:** Have you measured actual inference time improvements on different hardware configurations (e.g., CPU, embedded devices)?

- If inference time is not significantly reduced despite the smaller model size, the practical efficiency gains may be overstated. Reporting real-world latency measurements would strengthen the claim that LOP improves efficiency.

**Question 3:** Have you evaluated LOP’s compressed models on out-of-distribution (OOD) data or domain shifts? Does aggressive pruning increase sensitivity to dataset shifts?

- If LOP sacrifices robustness for compression, this would be a key limitation. A robustness analysis would clarify whether LOP is suitable for real-world deployment in dynamic environments.

**Question 4:** Did you consider comparing LOP against gradient boosting distillation or other distillation-based tree compression methods? If not, what are the key differences that make LOP preferable?

- If LOP outperforms both pruning-based and distillation-based compression techniques, it would further validate its significance. If not, distillation may be a viable alternative that should be acknowledged.

**Question 5:** How does the runtime of LOP compare to the time required to train an original XGBoost model? Is LOP feasible for large-scale datasets?

- If LOP takes significantly longer than training from scratch, its use case may be limited to scenarios where model compression is a strict requirement.

**Question 6:** Have you tested LOP on Random Forests, BART, or other tree ensemble models?

- If LOP is specifically optimized for boosting-based ensembles (e.g., XGBoost) but does not work as effectively on bagging-based methods like Random Forests or Bayesian ensemble methods like BART, its applicability may be more limited than suggested. Demonstrating strong performance across diverse tree-based models would increase confidence in LOP’s versatility.

**Relation To Broader Scientific Literature:**

LOP builds on prior research in tree ensemble pruning, model compression, and robustness verification by introducing a level-wise pruning approach that optimizes leaf values while reducing model size. Unlike traditional ensemble pruning methods (e.g., Margineantu & Dietterich, 1997; Tsoumakas et al., 2009) that remove entire trees, or global refinement techniques (Ren et al., 2015) that adjust leaf predictions post hoc, LOP prunes subtrees at any level while simultaneously updating leaf values, achieving significantly higher compression with minimal accuracy loss. Its focus on reducing memory footprint aligns with recent efforts to make tree ensembles more efficient for edge computing (Fan et al., 2013; Daghero et al., 2021). Additionally, by producing smaller models, LOP accelerates robustness verification (Kantchelian et al., 2016; Devos et al., 2021), addressing a key challenge in ensuring the safety and fairness of machine learning systems.

**Theoretical Claims:**

The paper does not present detailed formal proofs of correctness.

---

> ### Author Rebuttal · Authors · 2025-04-01
>
> Many thanks for your comments. We acknowledge that this work has many links with XAI (including knowledge distillation), and also with robustness, verification, inference efficiency, and other topics. However, the focus of this work is on the specific task of ensemble compression; we position this work in that area and that is also what the comparative evaluation focuses on. Reaching out to all these other areas would lengthen the paper, would raise more questions that warrant investigating, and would detract from the main message. Rejecting the paper just for not doing this would mean it is held to a much higher standard than related work in this area (none of that work does all these comparisons).
>
> Q1: LOP can be applied to regression by using an appropriate regression loss in Eq. 2. Applying it to multi-class problems would require a wrapper around it, such as one-versus-all classification. The same holds for most of the related work.
> For regression, we trained multiple XGBoost ensembles with [10, 25, 50, 100] trees and [4, 6, 8] max depth with learning rate 0.1 on 2 datasets (Wine Quality, Houses) using 5 fold CV. These datasets are a subset of those used in the paper by Liu et al. (2023) as suggested by reviewer CuzX. LOP achieves an average compression ratio of 14.3, compared to 7.7 for GR, 1.4 for LRL1 and 4.4 for ForestPrune (Liu et al., 2023). In terms of predictive performance, LOP increases the Root Mean Square Error by only 2% compared to the uncompressed model, which equals the allowed loss in performance (i.e., hyperparameter $\Delta$). Hence, the conclusions are similar to the binary classification case. We will add these results to the paper.
>
>
> Q2: We have not measured actual times; prediction requires sequential execution of if-statements and it seems obvious that the computation time is linear in the number of such statements.
>
> Q3: We have not investigated this; this is a different research question, completely out of scope for this paper. None of the related work investigates this, and we do not see how it could even fit in an 8 pages paper.
>
> Q4: As said, we focus on ensemble compression, and while exploring links with distillation based approaches may be interesting, we consider it out of scope in this paper. Furthermore, we do not know of distillation approaches that start from a forest and return a forest; if the reviewer has specific approaches in mind, a concrete reference would be appreciated.
>
> Q5: LOP typically takes longer than XGBoost itself. Ensembles are trained (and potentially compressed) once but used many times: the point of this work is to make them more efficiently usable / analyzable. The combination of training and compression time is usually in the order of seconds or minutes on a laptop: it is not an issue in practice.
>
> Q6: In response to the reviews, we ran experiments on Random Forests. Results are comparable to those with XGBoost.
> Specifically, we use Scikit-Learn Random Forest classifiers with a maximum tree depth in [10, 15] and number of trees in [100, 250, 500], on 3 representative datasets: Spambase, Phoneme and California. A full table of results can be found in our answer to reviewer N4NE (this table also contains results for an additional method called ForestPrune (Liu et al. 2023) suggested by reviewer CuzX). LOP averages a compression ratio of 228.2, whereas GR achieves a ratio of 120.2, and both IC and LRL1 achieve a ratio of 1.3. The average drop in balanced accuracy for LOP is 0.01, which is similar to most baselines. In terms of run time, it takes LOP on average 164 seconds to compress these ensembles, compared to 6892 seconds for GR, and 5071.8 seconds for LRL1. IC takes on average 71.6 seconds, but it has much worse compression ratios than LOP. Hence, LOP compresses more than the competitors without losing accuracy. Moreover, it is around 42x faster than GR which is its nearest competitor in terms of compression ratio. We will add these results to the paper.

---

### Decision · Program_Chairs · 2025-05-01

**Decision:**

Accept (poster)

**Comment:**

This paper proposes a method to compress ensembles of decision trees. With this goal, the authors propose LOP, a method for pruning ensemble members or entirely removing them from the ensemble. The leaf predictions are updated in such a way that predictive accuracy is mostly unaffected. The authors carry out several experiments showing that LOP gives compression factors that are often 10 to 1000 better than other competing methods. The reviewers indicate that experimental design in the paper appears well-structured, with a strong benchmarking approach against relevant baselines. A limitation is that the evaluation is only on binary classification problems, however. The reviewers also indicate that the paper is generally well written. Some reviewers expressed points of criticism regarding missing comparisons with related methods from the literature. However, the authors in the rebuttal provided extra experimnetal results comparing with some of them. Overall I believe this is a borderline paper, but I think it is closer to the acceptance level.